# ASSESSING MODEL OUT-OF-DISTRIBUTION GENERALIZATION WITH SOFTMAX PREDICTION PROBABILITY BASELINES AND A CORRELATION METHOD

## ABSTRACT

This paper studies the use of Softmax prediction probabilities to assess model generalization under distribution shift. Specifically, given an out-of-distribution (OOD) test set and a pool of classifiers, we aim to develop a prediction probability-based measure which has a monotonic relationship with OOD generalization performance. We first show existing uncertainty measures (*e.g.*, entropy and maximum Softmax prediction probability) are fairly useful of predicting generalization in some OOD scenarios. We then move ahead with proposing a new measure, Softmax Correlation (SoftmaxCorr). To obtain the SoftmaxCorr score for a classifier, we compute the class-class correlation matrix from all the Softmax vectors in a test set, and then its cosine similarity with an identity matrix. We show that the class-class correlation matrix reveals significant knowledge about the confusion matrix: its high similarity with the identity matrix means predictions have low confusion (uncertainty) and evenly cover all classes, and vice versa. Across three setups including ImageNet, CIFAR-10, and WILDS, we show that SoftmaxCorr is well predictive of model accuracy on both in-distribution and OOD datasets.

## 1 INTRODUCTION

Understanding the generalization of deep neural networks is an essential problem in deep learning. There is substantial interest in predicting ID generalization gap via complexity measures (Jiang et al., 2020a; Neyshabur et al., 2015; Bartlett et al., 2017; Keskar et al., 2017; Nagarajan & Kolter, 2019; Neyshabur et al., 2017; Chuang et al., 2021; Jiang et al., 2020b; Smith & Le, 2018; Arora et al., 2018; Dziugaite & Roy, 2017; Dinh et al., 2017). Although significant, developing measures to characterize OOD generalization remains under-explored. In fact, the test environment in the real world often undergoes distribution shift due to factors like sample bias and non-stationarity. Ignoring the potential distribution shift can lead to serious safety concerns in self-driving cars (Kuutti et al., 2020) and histopathology (Bandi et al., 2018), *etc*.

Softmax prediction probability is found to be useful to analyze the test environment (Hendrycks & Gimpel, 2016; Guillory et al., 2021; Deng et al., 2022a; Liang et al., 2018; Garg et al., 2022). For example, Hendrycks & Gimpel (2016); Liang et al. (2018) utilize maximum Softmax prediction probability to identify samples from open-set classes. This gives us a hint: models' prediction probabilities may be informative to reflect their OOD performance. Therefore, we are interested in measures based on prediction probability and conduct large-scale correlation study to explore whether they are useful to characterize generalization of various models under distribution shift. Concretely, given various deep models, we aim to study and develop prediction probability-based measures which monotonically relate to model generalization.

To this end, we construct a catalog of empirical prediction probability-based measures and create a wide range of experimental setups. We collect 502 different classification models ranging from standard convolutional neural network to Vision Transformers. We cover 19 ID and OOD datasets with various types of distribution shift, such as ImageNet-V2 (Recht et al., 2019) with dataset reproduction shift and stylized-ImageNet (Geirhos et al., 2019) with style shift.

Based on experimental results, we observe that empirical uncertainty measures based on prediction probabilities (*e.g.*, entropy) are useful in characterizing OOD generalization to some extent. How-

ever, they are limited in leveraging class-wise relationship encoded in prediction probabilities. We thus further propose Softmax correlation (SoftmaxCorr), an effective prediction probability-based measure describing for each classifier to what extent classes correlate with each other. Specifically, for each classifier we compute a class correlation matrix from all prediction vectors in a test set. Then, we calculate its cosine similarity with an identity matrix to evaluate whether this classifier makes diverse and certain predictions. We show that class-class correlation effectively uncovers knowledge of confusion matrix, thus better reflecting overall accuracy on OOD test set. The broad correlation study shows the efficacy of SoftmaxCorr. Our contributions are summarized below.

- We observe that Softmax prediction probability-based measures generally give good baseline indicators of the OOD accuracy of various classification models.

- Furthering this finding, we propose SoftmaxCorr which assesses model generalization by explicitly calculating class-class correlation, a new angle to leverage the prediction probability. A wide range of experiment suggests the effectiveness of this method.

## 2 RELATED WORK

**Predicting generalization in deep learning** studies the ID generalization gap (*i.e.*, the difference between training and test accuracy) of deep neural networks. Representative methods develop complexity measures based on model parameters and training set (Jiang et al., 2020a;b; Neyshabur et al., 2015; Keskar et al., 2017; Bartlett et al., 2017; Neyshabur et al., 2018; Liang et al., 2019; Chuang et al., 2021; Smith & Le, 2018; Arora et al., 2018; Dziugaite & Roy, 2017; Dinh et al., 2017), such as distance of training weights from initialization (Nagarajan & Kolter, 2019), the product of norms of weights across layers (Neyshabur et al., 2017) and the change of model accuracy with respect to different perturbation levels in training data (Schiff et al., 2021). These methods assume that training and test data come from the same distribution and do not incorporate characteristics of test data, so we can unlikely make reasonable predictions on a different distribution. To mitigate this limitation, we investigate the model generalization under distribution shift by developing measures that reflect models' generalization property on OOD datasets.

**OOD generalization.** Machine learning models should generalize from training distribution to OOD datasets (Djolonga et al., 2021; Koh et al., 2021). To study the problem of OOD generalization, several benchmarks are proposed (Hendrycks & Dietterich, 2019; Koh et al., 2021; Gulrajani & Lopez-Paz, 2021), such as corruption benchmark (Hendrycks & Dietterich, 2019) and domain generalization testbed (Gulrajani & Lopez-Paz, 2021). Moreover, several methods are proposed to improve model OOD generalization (Volpi et al., 2018; Zhao et al., 2020; Sagawa et al., 2020; Liu et al., 2021a; Mansilla et al., 2021; Shi et al., 2021), such as adversarial domain augmentation (Volpi et al., 2018; Qiao & Peng, 2021) and inter-domain gradient matching (Shi et al., 2021).

There are few works study the characterization of model OOD generalization. Ben-David et al. (2006; 2010) provide upper bounds of OOD generalization error for domain adaptation. Some works further bound the OOD generalization error based on the divergence between the two distributions (Acuna et al., 2021; Zhang et al., 2019; Tachet des Combes et al., 2020). However, as suggested by Miller et al. (2021), when the shift becomes larger, the above bounds on OOD performance become looser. Moreover, Vedantam et al. (2021) report that the adapting theory from domain adaptation is limited in predicting OOD generalization. In this work, we aim to assess model generalization under distribution shift by OOD measures.

**Out-of-distribution detection** aim to detect test data that do not belong to any of classes modeled in training distribution (Hendrycks & Gimpel, 2016; Yang et al., 2021a). Many methods are developed based on model outputs (Hendrycks & Gimpel, 2016; Liang et al., 2018; Hendrycks et al., 2022). For example, Hendrycks & Gimpel (2016) use maximum prediction probability to identify OOD samples. Hendrycks et al. (2022) further show that, without being normalized by Softmax function, maximal model logit is also a strong baseline for OOD detection. Liu et al. (2020b) calculate an energy score based on Softmax probability, which is regarded as a simple yet effective replacement for maximum prediction probability. Other methods investigate the predictive uncertainty (Liu et al., 2020a; Van Amersfoort et al., 2020). Different from the above works, this research does not aim to detect OOD data, but to explore Softmax-based measures to assess model generalization.

## 3 TASK FORMULATION

**Task definition.** We consider a $K$-way classification task, and let $\mathcal{Y} = \{1, ..., K\}$ denote the label space and $\mathcal{X} \in \mathbb{R}^d$ denote the input space. We are given a labeled training set $\mathcal{D}^S :=$ $\{(x_i^s, y_i^s)\}_{i=1}^{N_s}$ that contains $N_s$ data i.i.d drawn from a source distribution $P_S$, and an OOD test set $\mathcal{D}^T := \{(x_i, y_i)\}_{i=1}^N$ that contains $N$ data i.i.d drawn from another distribution $P_T$ ($P_S \neq P_T$). We train $M$ neural network classifiers $\{\phi_m\}_{m=1}^M$ on $\mathcal{D}^S$. Given a sample $(\boldsymbol{x}, y)$ from $\mathcal{D}^T$, the classifier $\phi_m : \mathcal{X} \to \Delta^K$ generates Softmax prediction probabilities for $\boldsymbol{x}$ on $K$ classes, where $\Delta^K$ denote $K - 1$ dimensional unit simplex. By testing on $\mathcal{D}^T$, $\phi_m$ yields a prediction matrix $\boldsymbol{P} \in \mathbb{R}^{N \times K}$, whose each row represents prediction probabilities of a test data. Specifically, the prediction matrix satisfies $\sum_{j=1}^k P_{i,j} = 1 \; \forall i \in 1 \dots N$ and $P_{i,j} \geq 0 \; \forall i \in 1 \dots N, j \in 1 \dots K$, where $P_{i,j}$ indicates the probability that $i$-th sample is predicted to the $j$-th class.

The dataset has an evaluation metric (*e.g.*, accuracy) to obtain ground-truth generalization $G_m$ of classifier $\phi_m$. The goal is to design a measure to calculate a score $S_m$ for each classifier $\phi_m$ without access to data annotations. The calculated scores $\{S_m\}_{m=1}^M$ ideally should well correlate with $\{G_m\}_{m=1}^M$, so that we can assess the OOD generalization of models based on the scores.

**Evaluation metrics.** We use Spearman's Rank Correlation coefficient $\rho$ (Kendall, 1948) to measure monotonicity between calculated scores and model generalization. We additionally use the weighted variant of Kendall's rank correlation $\tau_w$, which is shown to be useful when selecting the best ranked item is of interest (You et al., 2021). Both coefficients range from $[-1, 1]$. A value closer to $-1$ or $1$ indicates a strong negative or positive correlation, respectively, and $0$ means no correlation. Similar to Miller et al. (2021); Baek et al. (2022), we apply the same probit scale to both accuracy and SoftmaxCorr in our experiment for better linear fit.

## 4 SOFTMAX PREDICTION PROBABILITY-BASED OOD MEASURES

### 4.1 WHY USE SOFTMAX PREDICTION PROBABILITY?

Deep neural networks tend to make over-confident predictions (Ovadia et al., 2019; Minderer et al., 2021; Guo et al., 2017; Hein et al., 2019), which at the first look makes it less reliable to use Softmax Prediction as an uncertainty measure on the test data. However, existing works show that it is indeed informative in analyzing test environments. For example, Hendrycks & Gimpel (2016) show that the maximum Softmax prediction probability (MaxPred) of correct samples tends to be higher than that of incorrect or OOD samples. Guillory et al. (2021); Garg et al. (2022) suggest it is useful to estimate accuracy of a trained classifier by calculating its MaxPred on test samples.

**Proof of concept.** The above works imply the prediction probability is likely to be effective in measuring OOD performance of a pool of models. Given an OOD test set (ImageNet-R) and various ImageNet models, we conduct correlation study between MaxPred and classification accuracy. In Fig. 1, we show that there is a moderate correlation between MaxPred and model accuracy ($\rho = 0.613$ and $\tau_w = 0.776$) on ImageNet-R. This indicates that MaxPred is feasible in assessing OOD performance. Based on this observation, we further explore more empirical prediction probability-based measures and develop a more effective measure which exploits more semantics reflected in the prediction probability.

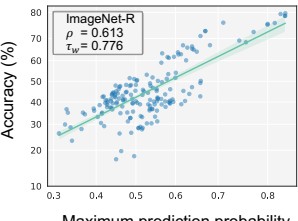

Figure 1: Correlation study between MaxPred and accuracy (%) on ImageNet-R.

### 4.2 EXPLORING MORE EMPIRICAL MEASURES

In addition to MaxPred, we investigate other empirical measures below.

**Softmax Gap** (SoftGap). MaxPred only utilizes the maximal value of Softmax vectors while disregarding values on other entries. We introduce SoftGap based on MaxPred which further considers the second largest entry in a prediction vector. Specifically, it calculates the average of difference between the largest and second largest Softmax prediction probabilities over all samples. A larger Softgap indicates prediction is more certain, suggesting better generalization.

Figure 2: Comparison between the correlation matrix computed from Softmax vectors and confusion matrix. We show comparisons obtained by four classifiers: LeNet, SimpleDLA and MobileNet. The test set is CINIC. For each pair of matrices, we use representational similarity analysis ($r_s$) to measure their similarity. The high $r_s > 0.85$ indicate that the class correlation matrix reveals knowledge of confusion matrix and therefore model accuracy.

**Negative Predictive Entropy** (Entropy) (Pereyra et al., 2017; Vedantam et al., 2021). This computes negative entropy of $\boldsymbol{P}$ by $-\frac{1}{N}\sum_{i=1}^{N} H(\boldsymbol{P}_{i,:})$, where $H(\cdot)$ is Shannon entropy (Shannon, 1948). A larger Entropy means a model makes more confident predictions.

**Information Maximization** (InfoMax). It is a loss function used in unsupervised domain adaptation algorithms (Shi & Sha, 2012; Liang et al., 2020). Its formula is $H(\frac{1}{N}\sum_{i=1}^{N} \boldsymbol{P}_{i,:}) - \frac{1}{N}\sum_{i=1}^{N} H(\boldsymbol{P}_{i,:})$. The idea is that a well-generalized model has diverse predictions over all samples (first term) while being certain on individuals (second term). Specifically, an accurate model tends to have a large entropy of the average prediction probability of all samples and a small entropy on each data point.

## 4.3  OUR METHOD: SOFTMAX CORRELATION (SOFTMAXCORR)

Given the prediction matrix $\boldsymbol{P} \in \mathbb{R}^{N \times K}$ predicted by $\phi_m$, a class correlation matrix $\boldsymbol{C} \in \mathbb{R}^{K \times K}$ can be computed by $\boldsymbol{C} = \boldsymbol{P}^{\mathsf{T}} \boldsymbol{P}$. An entry $C_{i,j}$ indicates the correlation between prediction probabilities of class $i$ and class $j$ over all samples, and is computed by $C_{i,j} = \sum_{n=1}^{N} P_{n,i} P_{n,j}$. We define the sum of diagonal entries of the correlation matrix as intra-class correlation (IntraCorr) and off-diagonal elements as inter-class correlation (InterCorr). Note that, the sum of $\boldsymbol{C}$ is $N$.

Based on class correlation matrix $\boldsymbol{C}$, we develop SoftmaxCorr to take into account the two characteristics. First, whether the model produces confident predictions and thus its computed class correlation matrix $\boldsymbol{C}$ has a high intra-class correlation. Second, whether the model gives diverse predictions where all classes are predicted. This detects the trivial model solution where all data are confidently predicted as the same class. To achieve this, we define SoftmaxCorr as the cosine similarity between the class-class correlation matrix $\boldsymbol{C}$ and an identity matrix $\boldsymbol{I}_K$: $cos(\boldsymbol{C}, \boldsymbol{I}) = \frac{P \cdot I}{\|\boldsymbol{C}\| \cdot \|\boldsymbol{I}\|}$. A higher similarity score means that model gives 1) high prediction certainty (intra-class correlation) and 2) high prediction diversity (the diagonal elements of class correlation matrix is uniformly distributed). We note that existing studies (Yang et al., 2021b; Wang & Isola, 2020; Asano et al., 2019) also report that the two characteristics are important for learning discriminative features.

In Fig. 2, we compare the computed class correlation matrices and ground-truth confusion matrices of LeNet (LeCun et al., 1998), SimpleDLA (Yu et al., 2018) and MobileNet (Howard et al., 2017) on CINIC. We use representational similarity analysis ($r_s$) (Kriegeskorte et al., 2008; Dwivedi & Roig, 2019) to calculate the similarity between each pair of class correlation and confusion matrices. We observe that $r_s$ scores are high ($r_s > 0.85$). This suggests that correlation matrix encodes critical and informative knowledge of the level of confusion between classes, thus directly relating to accuracy.

## 5  EXPERIMENT

In this section, we first describe the ImageNet, CIFAR-10 and WILDS setups. Then, we analyze the experiment results of prediction probability-based measures on three setups. After that, we compare SoftmaxCorr with other characterizations of correlation matrix. Moreover, we test whether SoftmaxCorr can model checkpoints along the training trajectory and assess models trained by different domain adaption algorithms. Lastly, we study the correlation between SoftmaxCorr and accuracy of

Table 1: Method comparison on ImageNet, CIFAR-10, WILDS and DomainNet. We compare our SoftmaxCorr with four empirical Softmax prediction probability-based measures: MaxPred, Soft-Gap, Entropy and InfoMax. We use Spearman's rank correlation ($\rho$) and weighted Kendall's correlation ($\tau_w$) for comparison. The highest correlation in each row is highlighted in **bold**. We show that our method is stable and yields the highest average correlation on ImageNet, CIFAR-10 and WILDS setup and competitive correlation on DomainNet setup.

| Setup | Dataset | MaxPred | | SoftGap | | Entropy | | InfoMax | | SoftmaxCorr | |
|---|---|---|---|---|---|---|---|---|---|---|---|
| | | $\rho$ | $\tau_w$ | $\rho$ | $\tau_w$ | $\rho$ | $\tau_w$ | $\rho$ | $\tau_w$ | $\rho$ | $\tau_w$ |
| ImageNet | ImageNet-Val | 0.542 | 0.515 | 0.665 | 0.568 | 0.142 | 0.286 | 0.145 | 0.288 | **0.919** | **0.752** |
| | ImageNet-V2-A | 0.645 | 0.574 | 0.747 | 0.622 | 0.217 | 0.327 | 0.222 | 0.330 | **0.922** | **0.792** |
| | ImageNet-V2-B | 0.506 | 0.483 | 0.636 | 0.554 | 0.109 | 0.256 | 0.113 | 0.266 | **0.912** | **0.785** |
| | ImageNet-V2-C | 0.577 | 0.528 | 0.699 | 0.594 | 0.169 | 0.303 | 0.173 | 0.306 | **0.917** | **0.795** |
| | ImageNet-R | 0.613 | 0.776 | 0.797 | 0.839 | 0.295 | 0.623 | 0.423 | 0.688 | **0.925** | **0.880** |
| | ImageNet-S | 0.784 | 0.798 | 0.839 | 0.825 | 0.471 | 0.652 | 0.614 | 0.730 | **0.875** | **0.883** |
| | Stylized-ImageNet | 0.580 | 0.609 | 0.661 | 0.656 | 0.233 | 0.431 | 0.512 | 0.613 | **0.828** | **0.834** |
| | ObjectNet | 0.912 | 0.814 | **0.934** | 0.848 | 0.820 | 0.735 | 0.836 | 0.761 | 0.927 | **0.897** |
| | ImageNet-Blur | 0.877 | 0.828 | 0.898 | 0.843 | 0.780 | 0.702 | 0.850 | 0.807 | **0.967** | **0.929** |
| | Average | 0.671 | 0.665 | 0.764 | 0.705 | 0.361 | 0.480 | 0.433 | 0.520 | **0.910** | **0.839** |
| CIFAR-10 | CIFAR-10-Val | 0.860 | 0.512 | **0.871** | **0.551** | 0.844 | 0.494 | 0.844 | 0.494 | 0.865 | 0.536 |
| | CIFAR-10.2 | 0.857 | 0.696 | 0.868 | **0.705** | 0.826 | 0.663 | 0.832 | 0.669 | **0.884** | 0.694 |
| | CINIC | 0.753 | 0.375 | 0.777 | 0.402 | 0.722 | 0.331 | 0.735 | 0.348 | **0.844** | **0.578** |
| | CIFAR-10-Noise | 0.032 | -0.082 | 0.041 | -0.077 | 0.035 | -0.090 | **0.896** | 0.750 | 0.813 | **0.771** |
| | Average | 0.626 | 0.375 | 0.639 | 0.395 | 0.607 | 0.350 | 0.827 | 0.565 | **0.851** | **0.645** |
| WILDS | iWildCam-ID | **0.972** | **0.889** | 0.969 | 0.887 | 0.967 | 0.885 | 0.960 | 0.810 | 0.945 | 0.792 |
| | iWildCam-OOD | 0.917 | 0.840 | **0.936** | **0.846** | 0.853 | 0.798 | 0.883 | 0.792 | 0.916 | 0.839 |
| | Camelyon17-ID | 0.847 | 0.718 | 0.847 | 0.718 | 0.728 | 0.611 | 0.833 | 0.653 | **0.893** | **0.746** |
| | Camelyon17-OOD | 0.192 | 0.320 | 0.192 | 0.320 | 0.167 | 0.299 | **0.695** | **0.635** | 0.689 | 0.492 |
| | Average | 0.732 | 0.691 | 0.736 | 0.692 | 0.679 | 0.647 | 0.842 | **0.725** | **0.861** | 0.717 |
| DomainNet | DomainNet-ID | 0.725 | 0.570 | 0.744 | 0.597 | 0.677 | 0.523 | 0.698 | 0.539 | **0.747** | **0.607** |
| | DomainNet-OOD | 0.403 | 0.274 | 0.407 | 0.258 | 0.386 | 0.253 | **0.763** | **0.721** | 0.693 | 0.566 |
| | Average | 0.564 | 0.422 | 0.576 | 0.428 | 0.532 | 0.388 | **0.731** | **0.630** | 0.720 | 0.587 |

a single model on various OOD test sets. We note that for the completeness of experiment, we also include the performance of Softmax measures on ID test sets.

## 5.1 EXPERIMENTAL SETUP

**ImageNet setup.** We collect 173 models publicly accessible from TIMM (Wightman, 2019). They are trained or fine-tuned on ImageNet (Deng et al., 2009) and vary in architectures and training strategies. We use both ID and OOD datasets for correlation study. Specifically, ImageNet-Val is ID test set. OOD datasets are: (1) ImageNet-V2 (Recht et al., 2019); (2) ObjectNet (Barbu et al., 2019); (3) ImageNet-S(ketch) (Wang et al., 2019); (4) ImageNet-Blur severity 5 (Hendrycks & Dietterich, 2019); (5) ImageNet-R(endition) (Hendrycks et al., 2021); (6) Stylized-ImageNet (Geirhos et al., 2019). Note that, ImageNet-V2 has three versions (ImageNet-V2-A/B/C) resulting from different sampling strategies. ImageNet-R and ObjectNet contain 200 and 113 ImageNet classes respectively.

**CIFAR-10 setup.** We collect 65 networks trained with the scheme provided by Wightman (2017) on CIFAR-10 training set (Krizhevsky et al., 2009). CIFAR-10-Val(idation) is the ID test set. For OOD datasets, we use (1) CIFAR-10.2 (Recht et al., 2018) (2) CINIC (Darlow et al., 2018) (3) CIFAR-10-Noise with severity 5 (Hendrycks & Dietterich, 2019).

**WILDS setup.** We follow the same training scheme provided by Koh et al. (2021) to train or fine-tune models for Camelyon17 (Bandi et al., 2018) and iWildCam (Beery et al., 2021). **Camelyon17** is a binary classification dataset where the objective is to classify whether a slide contains tumor issue. We use 45 models varying in architectures and random seeds. **iWildCam** is a 182-way

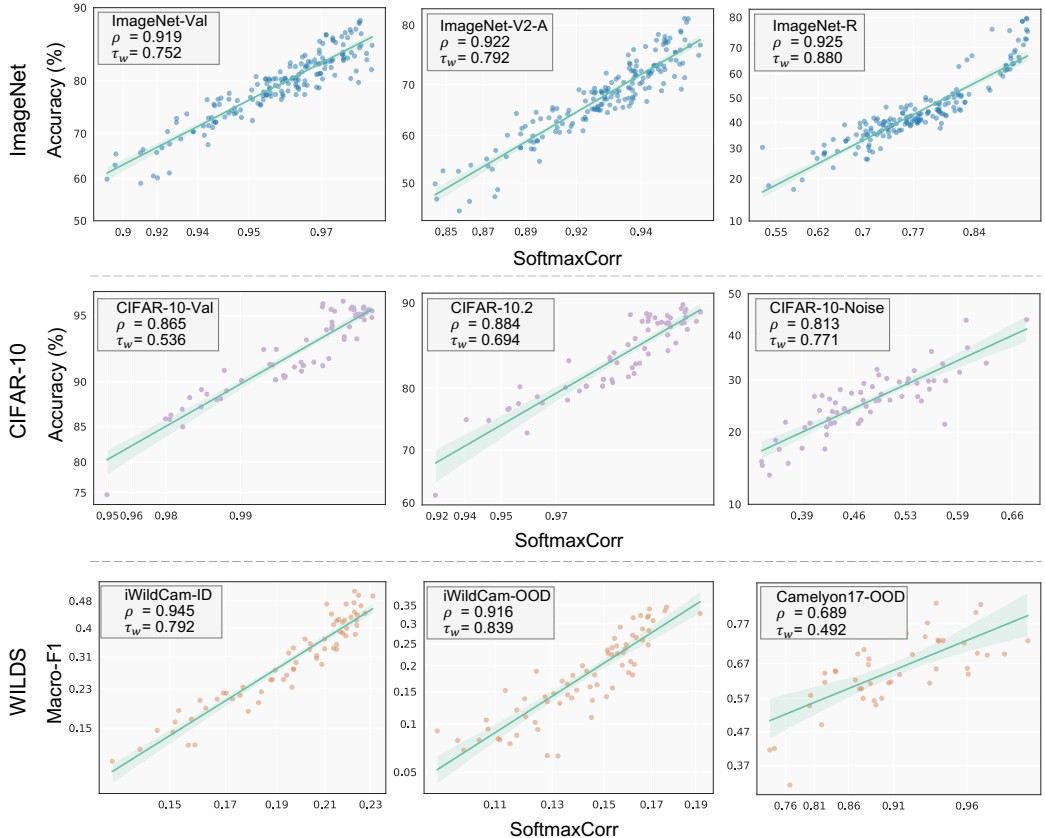

Figure 3: SoftmaxCorr *vs.* model generalization under ImageNet, CIFAR-10 and WILDS setups. For ImageNet setup, ID test set is ImageNet-Val; OOD test sets are ImageNet-V2-A and ImageNet-R. For CIFAR-10 setup, ID test set is CIFAR-10-Val; OOD test sets are CIFAR-10.2 and CIFAR-10-Noise. For WILDS, ID test set is iWildCam-ID; OOD test sets are iWildCam-OOD and Camelyon17-OOD. The $y$-axis is top-1 accuracy, top-1 accuracy and macro-F1 for the three setups, respectively. In each subfigure, each point denotes a model trained for the corresponding task. Straight lines are fit with robust linear regression (Huber, 2011). Axes are probit scaled as described in Section 3. We observe that SoftmaxCorr is a reliable and effective metric. Particularly on ImageNet, SoftmaxCorr is predictive of model generalization with strong performance ($\rho > 0.92$).

animal classification dataset. We collect 66 models the variations of which results from different network architectures and learning rates. Model performance is measured by macro-$F1$ score.

## 5.2 MAIN OBSERVATIONS

**SoftmaxCorr exhibits a strong correlation with model generalization.** In Fig. 3 and Table 1, we observe that SoftmaxCorr is indicative of model performance under the three setups. Particularly on the ImageNet setup, SoftmaxCorr has consistently stronger correlations than the other empirical measures. For example, the average Spearman's Rank Correlation coefficient $\rho$ is $0.910$, $0.671$ and $0.764$, and $0.361$, and $0.433$ for our method, MaxPred, SoftGap, Entropy and InfoMax, respectively. On CIFAR-10 and WILDS, while on some test sets it does not present the strongest correlation, we still think that SoftmaxCorr is a preferred measure because it has higher or very competitive average correlation scores ($\rho = 0.851, 0.861$ and $\tau_w = 0.645, 0.717$).

**SoftmaxCorr gives more stable correlation, while the other four measures have mixed performance on different test sets.** On CIFAR-10-Noise, we find that SoftmaxCorr correlates well with model performance ($\rho = 0.813$ and $\tau_w = 0.771$). In contrast, Entropy, MaxPred and Soft-Gap show no correlation ($\rho < 0.05$ and $\tau_w < 0$). Although InfoMax has higher rank correlation

| Dataset | Diag-sum | Diag-std | SoftmaxCorr |
|---------|----------|----------|-------------|
| ImageNet-V2-A | 0.503 | 0.099 | **0.922** |
| ImageNet-R | 0.669 | 0.314 | **0.925** |
| CIFAR-10.2 | 0.854 | 0.688 | **0.884** |
| CINIC | 0.752 | 0.781 | **0.844** |
| iWildCam-OOD | **0.916** | -0.696 | **0.916** |
| Camelyon17-OOD | 0.174 | 0.581 | **0.689** |
| average | 0.645 | 0.294 | **0.863** |

Table 2: Comparison between SoftmaxCorr and two variants (Diag-sum and Diag-std). Rank correlation ($\rho$) is used as the metric.

| Dataset | 1% | 5% | 10% | 30% | 100% |
|---------|-----|-----|-----|-----|------|
| ImageNet-V2-A | 0.820 | 0.853 | 0.865 | 0.901 | 0.922 |
| ImageNet-R | 0.873 | 0.928 | 0.925 | 0.926 | 0.925 |
| CIFAR-Noise | 0.734 | 0.785 | 0.814 | 0.813 | 0.813 |
| CINIC | 0.655 | 0.777 | 0.827 | 0.857 | 0.844 |
| iWildCam-OOD | 0.826 | 0.874 | 0.896 | 0.903 | 0.916 |
| Camelyon17-OOD | 0.708 | 0.701 | 0.690 | 0.694 | 0.689 |

Table 3: Sensitivity analysis of SoftmaxCorr on test set sizes. We test four sampling ratios and report $\rho$ on six datasets. We show that SoftmaxCorr is relatively stable given a reasonable number of samples.

than SoftmaxCorr ($\rho = 0.896$ *vs.* 0.813), it does not correlate with accuracy on ImageNet-Val, ImageNet-V2-A/B/C ($\rho < 0.25$ and $\tau_w < 0.35$). On ID and OOD sets of iWildCam, SoftGap and MaxPred have higher correlations than SoftmaxCorr. However, they fail to measure the macro-F1 on Camelyon17-OOD ($\rho = 0.192$). While in some cases SoftmaxCorr does not present the highest correlation, we emphasize that it overall gives more stable and stronger correlations. Thus, we think that SoftmaxCorr is generally more indicative of model generalization than other measures.

**Compared to SoftGap and InfoMax, SoftmaxCorr better utilizes Softmax prediction probabilities.** We use SoftGap on top of MaxPred as a simple approach to explicitly consider more entries (the second largest probability) in Softmax predictions. Based on the entropy, InfoMax further considers the prediction diversity (its first term) and thereby it also leverages more information of Softmax predictions. As shown in Table 1, both methods achieve higher correlation results than MaxPred. This indicates that using more information of Softmax predictions is useful. Compared with SoftGap and InfoMax, the class-wise correlation considered in our SoftmaxCorr better reveals the knowledge encoded by Softmax predictions. This is supported by the higher average correlation and more stable performance of SoftmaxCorr under three setups.

### 5.3 DISCUSSION ON THE CHARACTERIZATION OF CORRELATION MATRIX

To investigate the importance of prediction diversity and certainty for predicting OOD generalization, we compare SoftmaxCorr with two variants: (1) the sum of diagonal entries in the class correlation matrix (Diag-sum); (2) the negative of standard deviation of diagonal elements in the class correlation matrix (Diag-std). In Table 2, we present Spearman's rank correlation of three methods on six datasets from three setups. We see that SoftmaxCorr is more predictive of OOD generalization than Diag-sum and Diag-std ($\rho = 0.863$ *vs.* 0.645 *vs.* 0.294). This means that it is important to measure both prediction diversity and certainty for OOD generalization assessment.

### 5.4 SENSITIVITY ANALYSIS ON TEST SET SIZE

We study the sensitivity of SoftmaxCorr to test set size. Specifically, we reduce the dataset size by randomly sampling $1\%, 5\%, 10\%$ and $30\%$ of the original data. We report the averaged Spearman's correlation of three random runs on six datasets (*e.g.*, ImageNet-V2-A, ImageNet-R, CINIC and iWildCam-OOD). As shown in Table 3, we observe that when the number of test data is very small ($1\%$), the correlation of SoftmaxCorr drops. When the dataset size increases ($\geq 5\%$), SoftmaxCorr exhibits stable and high correlation with model performance. This suggests that SoftmaxCorr requires a reasonable number of samples to capture the model OOD generalization.

### 5.5 EVALUATING GENERALIZATION ALONG A TRAINING TRAJECTORY

In previous sections, SoftmaxCorr is utilized to measure the performance of models varying in different architectures and training strategies. In practice, we are sometimes interested in evaluating the models at different training check points. Hence, we also analyze whether SoftmaxCorr is helpful in this case. We collect prediction probabilities of CINIC every 10 epochs along the training process of ResNet-20, DenseNet-121 (Huang et al., 2017), VGG11 (Simonyan & Zisserman, 2014) and MobileNet (Howard et al., 2017) trained on the CIFAR-10. In Fig. 4, we observe that SoftmaxCorr

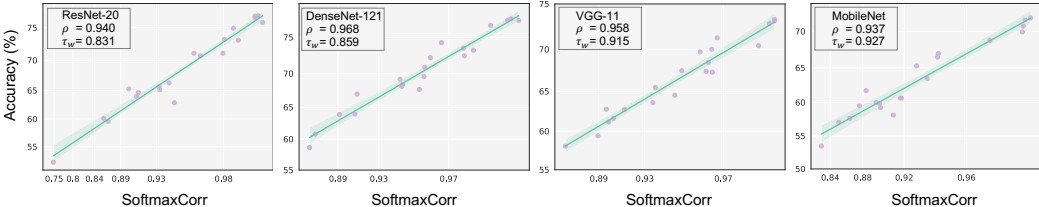

Figure 4: Correlation analysis: SoftmaxCorr and accuracy on CINIC. Each point represents a checkpoint. We consider CIFAR-10 models: ResNet-20, DenseNet-121, VGG-11 and MobileNet. Axes are probit scaled as in Section 3. In each subfigure, every point means a checkpoint of the model along the training process. For four models, we see strong correlations ($\rho > 0.93$). This suggests that SoftmaxCorr is helpful in assessing checkpoints along the training process.

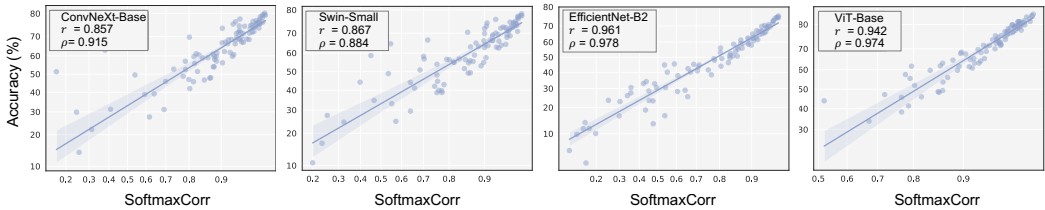

Figure 5: SoftmaxCorr *v.s.* accuracy on ImageNet-C benchmark. In every subfigure, each dot indicates a dataset of ImageNet-C. We observe strong linear and rank correlation for ImageNet models: ConvNext-Base, Swin-Small, EfficientNet-B2 and ViT-Base.

has high rank correlation ($\rho > 0.93$) with model performance for four networks. This means that we can potentially apply SoftmaxCorr to assay model generalization along the training process.

## 5.6  ASSESSING PREDICTIVE ABILITY OF SOFTMAXCORR UNDER DOMAIN ADAPTATION.

On ImageNet, CIFAR and WILDS setups, all models are trained by standard empirical risk minimization and do not use the unlabeled OOD samples for training. In some scenarios, domain adaptation (DA) algorithms are employed for learning target-adaptive models with additional unlabeled OOD samples (Kouw & Loog, 2019; Zhou et al., 2022). To explore whether SoftmaxCorr is still effective to assess the generalization of these models, we conduct correlation study under DomainNet setup (Peng et al., 2019; Sagawa et al., 2021). The models are trained by 9 different DA algorithms (*e.g.*, DeepCORAL (Sun & Saenko, 2016), DANN (Ganin et al., 2016)). In Table 1, we observe that SoftmaxCorr performs reasonably on DomainNet-ID and DomainNet-OOD. We also notice that InfoMax is better than SoftmaxCorr on DomainNet-OOD. InfoMax is commonly used as a regularization in DA (Shi & Sha, 2012; Liang et al., 2020). We speculate it might better reflect models' target-adaptation ability and exhibit higher correlation on DomainNet-OOD.

## 5.7  SOFTMAXCORR V.S. GENERALIZATION ON VARIOUS OOD TEST SETS

We investigate how a given trained model generalize to various OOD datasets. In detail, we evaluate a single model on all test sets of ImageNet-C benchmark (Hendrycks & Dietterich, 2019) and conduct correlation study between accuracy and SoftmaxCorr. We additionally use Pearson's correlation ($r$) to measure overall linear trend. This coefficient varies in $[-1, 1]$. A value closer to $-1/1$ indicates better negative/positive linearity and 0 means no correlation. We use ImageNet networks: ConvNeXt-Base (Liu et al., 2022), Swin-Small (Liu et al., 2021b), EfficientNet-B2 (Tan & Le, 2019) and ViT-Base (Dosovitskiy et al., 2020). Figure 5 shows a strong linear relationship and high rank correlation ($r > 0.85$ and $\rho > 0.91$). It indicates that with a linear regressor SoftmaxCorr can also help estimate the accuracy of a given model on various test sets.

## 6 DISCUSSION AND POTENTIAL DIRECTIONS

**What makes OOD measures interesting?** Miller et al. (2021) report an accuracy-on-the-line phenomenon where there exists a very strong linear correlation between probit-scaled ID and OOD generalization. It means ID generalization is a good predictor of OOD generalization. There are three reasons that make it meaningful to investigate OOD measures. **First**, *Softmax prediction probability can be obtained with just unlabeled data.* This makes prediction probability-based measures of important practical value as indicators of OOD generalization. **Second**, Miller et al. (2021) show that accuracy-on-the-line is not universal. That is, on some datasets, ID and OOD generalization do not show a clear positive correlation. This finding is further discussed in recent work (Wenzel et al., 2022). Specifically, Wenzel et al. (2022) suggest two patterns preventing this phenomenon: (1) underspecification (*e.g.*, Camelyon17) where same ID performance leads to different OOD behavior; (2) models do not transfer information from ID to OOD domains (*e.g.*, DomainNet). That is, despite of various ID performance, all models perform poorly on OOD datasets. Paradoxically, identifying the patterns itself requires labeled OOD data. Therefore, while ID performance is indicative in some scenarios, we think that SoftmaxCorr is overall a good alternative when labeled ID data is inaccessible or accuracy-on-the-line does not hold. **Third**, it is demanding and sophisticated to design ID test sets (Engstrom et al., 2020), which are expected to be unbiased and representative of distribution to effectively measure model ID accuracy. Further, it is a trade-off to split a full dataset into training, validation and test sets in terms of training and evaluation quality. **Last**, since SoftmaxCorr requires no held-out validation set, we can use all available data for training models.

**Discussion on imbalanced test sets.** SoftmaxCorr is defined as the cosine similarity between correlation matrix $C$ and an identity matrix $I_K$. This implicitly assumes the test set is balanced: the number of samples in each class is roughly equal. This definition may not optimal when the test set is extremely imbalanced (*e.g.*, some classes may only have one image being predicted into). For this point, we note that some test sets in our study are moderately imbalanced, such as ImageNet-R, ObjectNet, iWildCam and DomainNet. We empirically observe that SoftmaxCorr still maintains a high correlation on these datasets. Under extremely imbalanced case, the definition needs to consider the importance weighting of each class. It would be helpful to use techniques of label shift estimation (Lipton et al., 2018; Garg et al., 2020; Sun et al., 2022), and we leave it as our future work.

**Potential OOD measures.** This work proposes SoftmaxCorr to use class-wise relationship encoded by Softmax prediction probabilities. Here, we discuss other potential ways. **First**, SoftGap computes the difference of largest and second largest prediction probabilities. We show that SoftGap exhibits a stronger correlation with performance compared to MaxPred. It would be interesting to improve SoftGap by utilizing more probabilities (*e.g.*, top five probabilities). **Second**, for a perfectly calibrated model, its MaxPred over a test set corresponds to its accuracy. Yet, calibration methods seldom exhibit desired calibration performance under the distribution shift (Ovadia et al., 2019). That said, it would be promising to study post-hoc calibration methods for OOD datasets, which benefits MaxPred for assessing model generalization. **Third**, in some application scenarios, we might able to obtain a small number of labeled test data. It would be helpful to study the way of using them to improve the OOD generalization assessment. Last, this work focuses on Softmax prediction probability. We tested our method based on logits but no obvious correlation is exhibited. This may be because the logits of different models vary in significantly different ranges. Moreover, studying other model statistics (*e.g.*, weights and feature representations) would be interesting.

## 7 CONCLUSION

This paper studies an under-explored problem of assaying model generalization under distribution shift. To this end, we explore the use of Softmax prediction probability for developing OOD measures. We start by finding that maximum Softmax prediction probability is to some extent useful to measure the OOD performance. We then propose Softmax Correlation (SoftmaxCorr) which leverages class confusion encoded by class-class correlation matrix and thus better reflects the overall quality of the classifier predictions. To validate the usefulness of SoftmaxCorr, we compare with four other empirical measures across 19 datasets under ImageNet, CIFAR-10 and WILDS setups. We observe SoftmaxCorr generally presents more stable and higher correlation with model generalization on both ID and OOD datasets. This paper establishes some baseline usage of Softmax prediction probability and a specific improvement, and more investigation will be made in future.

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

# A  APPENDIX

## A.1  EXPERIMENTAL SETUP

### A.1.1  IMAGENET SETUP

**(1) ImageNet Models**

There are 173 models used in experiment. All of models are trained or fine-tuned on ImageNet-1k (Deng et al., 2009). The weights of the models are publicly avaiable via TIMM-1.5 (Wightman, 2019). For the choice of the models, we refer to the supplementary materials of Deng et al. (2022b); Miller et al. (2021); Baek et al. (2022). Moreover, we additionally include following models: { *'resnet10t', 'resnet14t', 'darknet53', 'cs3darknet_m', 'cs3darknet_focus_m', 'cs3darknet_l', 'cs3darknet_focus_l', 'regnety_040', 'regnety_064', 'regnety_080', 'regnetv_040', 'regnetv_064', 'regnetz_040', 'regnetz_040h'* }

**(2) ImageNet Datasets**

The datasets we use are publicly available standard benchmarks. We have double checked their license. We list their open-source as follows:

**ImageNet-Validation** (Deng et al., 2009) (https://www.image-net.org);
**ImageNet-V2-A/B/C** (Recht et al., 2019) (https://github.com/modestyachts/ImageNetV2);
**ImageNet-Corruption** (Hendrycks & Dietterich, 2019) (https://github.com/hendrycks/robustness);
**ImageNet-Sketch** (Wang et al., 2019) (https://github.com/HaohanWang/ImageNet-Sketch);
**ImageNet-Rendition** (Hendrycks et al., 2021) (https://github.com/hendrycks/imagenet-r);
**ObjectNet** (Barbu et al., 2019) (https://objectnet.dev);
**Stylized-ImageNet** (Geirhos et al., 2019) (https://github.com/rgeirhos/Stylized-ImageNet).

### A.1.2  CIFAR-10 SETUP

**(1) CIFAR-10 Models**

Follow the practice in (Miller et al., 2021), we train 65 CIFAR models using the implementations from https: //github.com/kuangliu/pytorch-cifar. The models span a range of manually designed architectures and the results of automated architecture searches. Specifically, we use: {*'DenseNet-121/169/201/161/201', 'Densenet-cifar', 'DLA', 'DPN26/92', 'EfficientNetB0', 'GoogLeNet', 'LeNet', 'MobileNet', 'MobileNetV2', 'PNASNetA/B', 'PreActResNet18/34/50/101/152', 'RegNetX-200MF/400MF', 'RegNetY-400MF', 'ResNet-18/34/50/101/152', 'ResNeXt29-8x64d/32x4d/4x64d/2x64d', 'SENet18', 'ShuffleNetV2', 'ShuffleNetG2/G3', 'SimpleDLA', 'VGG-11/13/16/19'*}.

Furthermore, we use the trained models publicly provided by https://github.com/chenyaofo/pytorch-cifar-models. They are {*'cifar10-mobilenetv2-x0-5', 'cifar10-mobilenetv2-x0-75', 'cifar10-mobilenetv2-x1-0', 'cifar10-mobilenetv2-x1-4', 'cifar10-repvgg-a0', 'cifar10-repvgg-a1', 'cifar10-repvgg-a2', 'cifar10-resnet20', 'cifar10-resnet32', 'cifar10-resnet44', 'cifar10-resnet56', 'cifar10-shufflenetv2-x0-5', 'cifar10-shufflenetv2-x1-0', 'cifar10-shufflenetv2-x1-5', 'cifar10-shufflenetv2-x2-0', 'cifar10-vgg11-bn', 'cifar10-vgg13-bn', 'cifar10-vgg16-bn', 'cifar10-vgg19-bn'*} .

**(2) CIFAR-10 Datasets**

The datasets are publicly available benchmarks. Their source are as follows:

**CIFAR-10** (Krizhevsky et al., 2009) (https://www.cs.toronto.edu/ kriz/cifar.html);
**CIFAR-10-C** (Hendrycks & Dietterich, 2019) (https://github.com/hendrycks/robustness);
**CIFAR-10.1** (Recht et al., 2018) (https://github.com/modestyachts/CIFAR-10.1);
**CINIC** (Darlow et al., 2018) (https://github.com/BayesWatch/cinic-10).

### A.1.3  WILDS SETUP

**(1) iWildCam Models**

We use the publicly available code (https://github.com/p-lambda/wilds) to finetune 66 models that are pretrained on ImageNet. Following the practice in (Miller et al., 2021), model variation results

from model architecture and learning rate. We train for 12 epochs with batch size 16 using Adam and sweep over learning rate in the grid $\{1e^{-3}, 1e^{-2}\}$. The other Adam parameters were set to the Pytorch defaults. The models are: { *'alexnet', 'mobilenet_v2', 'resnet18', 'resnet34', 'resnet50', 'vgg11', 'densenet121', efficientne_b0, efficientnet_b1, 'regnet_y_400mf', 'regnet_y_800mf', 'shufflenet_v2_x0_5' 'shufflenet_v2_x1_0'* }.

**(2) Camelyon17 Models**

We use the publicly available code (https://github.com/p-lambda/wilds) to finetune 45 models that are pretrained on ImageNet. Models vary in model architecture and random seed. The model architectures are: { *'alexnet', 'vgg11', 'densenet121', 'resnet18', 'resnet34', 'resnet50', 'resnet101', 'resnext50_32x4d' 'wide_resnet50_2' 'mobilenet_v3_large', 'regnet_y_1_6gf', 'shufflenet_v2_x0_5', 'shufflenet_v2_x1_5', 'shufflenet_v2_x2_0', 'regnet_y_400mf', 'regnet_y_800mf',* }.

**(3) WILDS Datasets**

iWildCam (Beery et al., 2021) https://www.kaggle.com/c/iwildcam-2020-fgvc7;
Camelyon17 (Bandi et al., 2018) https://camelyon17.grand-challenge.org/

### A.1.4    DomainNet

**Domain adaptation algorithms.**    We use publicly accessible weights provided in: https://worksheets.codalab.org/worksheets/0x52cea64d1d3f4fa89de326b4e31aa50a. These models are trained by following optimization algorithms: ERM (Vapnik, 1991), DeepCORAL (Sun & Saenko, 2016), DANN (Ganin et al., 2016), AFN (Xu et al., 2019), PseudoLabel (Lee et al., 2013), NoisyStudent (Xie et al., 2020), FixMatch (Sohn et al., 2020), SwAV (Caron et al., 2020).

**DomainNet Dataset** (Peng et al., 2019) http://ai.bu.edu/M3SDA/.

### A.2    Computation Resources

All experiment is run on one 2080Ti and the CPU AMD Ryzen Threadripper 2950X 16-Core Processor. We use PyTorch (1.10.2+cu102) for all the experiments.

### A.3    Discussion on low performance of SoftmaxCorr on Camelyon17-OOD

Camelyon17 is a binary image classification task where each example is a tissue patch. Its images are metastasized breast cancer tissue samples collected from different hospitals. As noted by Koh et al. (2021), the held-out OOD test set contains tissue samples from a hospital not seen in the training set. The distribution shift largely arises from differences in staining and imaging protocols across hospitals.

In our study, we observe a similar underspecification phenomenon as Miller et al. (2021); Wenzel et al. (2022): very similar ID performances lead to very different OOD performances. This indicates that the models cannot learn robust patterns that can be transferred to OOD data. Moreover, the similarity between images from the same slide or hospital is very high. This probably exacerbates the underspecification. Based on the above, we speculate that Softmaxcorr is less effective in detecting the underspecification issue on Camelyon17-OOD and thus has low correlation. That said, it would be interesting to further study this phenomenon in future research.

### A.4    Compared with more baselines

We additionally add three baselines: MaxLogit(Hendrycks et al., 2022), Energy(Liu et al., 2020b) and Adversarial Input Margin(Baldock et al., 2021).

**MaxLogits** calculates the average of maximal unnormalized logits of all test samples.

**Energy** calculates the average of energy scores of all samples. Its formula for a single input $x$ is $E = -T \cdot \log \sum_j^K e^{\phi^j(\mathbf{x})/T}$, where $T$ is the temperature scaling hyper-parameter and $\phi^j(x)$ indicates the logit returned by $\phi(x)$ corresponding to $j^{th}$ class.

Table 4: Method comparison on three OOD test sets. We compare our SoftmaxCorr with three measures: MaxLogit, Energy and Complexity. We use Spearman's rank correlation ($\rho$) and weighted Kendall's correlation ($\tau_w$) for comparison. We show that our method is stable and yields the highest average correlation across four datasets.

| Dataset | MaxLogit | | Energy | | Complexity | | SoftmaxCorr | |
|---|---|---|---|---|---|---|---|---|
| | $\rho$ | $\tau_w$ | $\rho$ | $\tau_w$ | $\rho$ | $\tau_w$ | $\rho$ | $\tau_w$ |
| ImageNet-V2-A | -0.267 | -0.237 | 0.350 | 0.158 | 0.341 | 0.282 | 0.922 | 0.792 |
| ImageNet-S | 0.024 | 0.422 | 0.087 | -0.180 | 0.329 | 0.417 | 0.875 | 0.883 |
| CIFAR-10.2 | 0.210 | -0.032 | -0.180 | -0.180 | 0.680 | 0.524 | 0.884 | 0.694 |

**Adversarial Input Margin** (Complexity) is proposed to measure the model complexity. It computes the smallest norm required for an adversarial perturbation in the input to change the model's class predictions. It estimates adversarial input margin $\gamma$ by a linear approximation. Its formula is given by: $\gamma \simeq \min_{j \neq i} \frac{|z_i - z_j|}{|\nabla_x(z_i - z_j)|}$, where $x$ is the input, $i$ means predicted class and $z_j$ denotes the logit returned by the network for class $j$. We then take the average of $\gamma$ over all data.

As shown in Table 4, we observe that MaxLogit and Energy exhibit weak correlation. While model complexity achieves some correlation on three datasets, it is still inferior to SoftmaxCorr.

## A.5 COMPARE SOFTMAXCORR WITH ACCURACY-ON-THE-LINE

We report the Spearman's rank correlations of accuracy-on-the-line and SoftmaxCorr in Table 5. We observe that accuracy-on-the-line shows strong correlations on many datasets. However, we have similar observations as the authors of accuracy-on-the-line report: it exhibits weak correlation or even no correlation on CIFAR-10-Noise, Camelyon17-OOD and DomainNet-OOD ($\rho = 0.003$, $-0.021$ and $0.350$). In comparison, SoftmaxCorr remains relatively informative and exhibits moderately high correlation ($\rho = 0.813$, $0.689$ and $0.693$) on the three test sets.

We further note that when accuracy-on-the-line shows very strong correlations, SoftmaxCorr is also competitive. Therefore, we think that SoftmaxCorr is overall a good alternative when labeled ID data is inaccessible or accuracy-on-the-line does not hold.

## A.6 AN ORACLE BASELINE: CONFUSION MATRIX

We may label a small number of OOD test data and compute the confusion matrix. Then, we calculate the cosine similarity between the computed confusion matrix and an identity matrix to assess model OOD generalization on the whole dataset. Specifically, we randomly label 1%, 5% and 10% of data and compute the cosine similarity between the confusion matrix and identity matrix. We report the average of Spearman's rank correlation over three random runs in Table 6

This oracle exhibits a strong correlation given 5% of labeled test data. This suggests that exploring class confusion information for OOD generalization assessment is feasible. Specifically, if the calculated class-class correlation matrix ideally matches the true confusion matrix, then OOD generalization can be perfectly predicted. Yet, we would like to discuss that this oracle may be impractical in some scenarios. First, data annotation could be very expensive and challenging. For example, precisely labeling wildlife categorization of iWildCam images is laborious and requires expert-level knowledge; annotating cancer tissue images of Camelyon17 also requires expert-level knowledge. Second, when the test distribution is changed, we need to label data again which is also laborious.

## A.7 SENSITIVITY ANALYSIS ON TEST SET SIZE OF ALL METHODS

We conducted a stratified analysis of all five methods on four OOD datasets: ImageNet-R, ImageNet-Blur, CINIC and iWildCam-OOD. Specifically, we test different size of OOD datasets by randomly sampling 1%, 5%, 10%, 30% of test data. For each method under each test set size, we report the Spearman's rank correlation ($\rho$) of the average results of three random runs.

| Dataset | ID-Acc | SoftmaxCorr |
|---|---|---|
| ImageNet-R | 0.932 | 0.925 |
| ImageNet-v2-A | 0.995 | 0.922 |
| ImageNet-v2-B | 0.994 | 0.912 |
| ImageNet-v2-C | 0.995 | 0.917 |
| ImageNet-S | 0.934 | 0.875 |
| Stylized-ImageNet | 0.800 | 0.828 |
| ObjectNet | 0.976 | 0.927 |
| ImageNet-Blur | 0.902 | 0.967 |
| CIFAR-10.2 | 0.958 | 0.884 |
| CINIC | 0.917 | 0.844 |
| CIFAR-10-Noise | 0.003 | 0.813 |
| iWildCam-OOD | 0.944 | 0.916 |
| Camelyon17-OOD | -0.021 | 0.689 |
| DomainNet-OOD | 0.350 | 0.693 |

Table 5: Comparison between Accuracy-on-the-line (ID-Acc) and SoftmaxCorr. Rank correlation ($\rho$) is the metric.

| Dataset | 1% | 5% | 10% |
|---|---|---|---|
| ImageNet-R | 0.984 | 0.993 | 0.993 |
| ImageNet-v2-A | 0.947 | 0.981 | 0.990 |
| ImageNet-v2-B | 0.857 | 0.956 | 0.974 |
| ImageNet-v2-C | 0.922 | 0.979 | 0.986 |
| ImageNet-S | 0.990 | 0.997 | 0.998 |
| Stylized-ImageNet | 0.986 | 0.996 | 0.998 |
| ObjectNet | 0.972 | 0.981 | 0.984 |
| ImageNet-Blur | 0.989 | 0.993 | 0.995 |
| CIFAR-10.2 | 0.350 | 0.617 | 0.706 |
| CINIC | 0.842 | 0.928 | 0.943 |
| CIFAR-10-Noise | 0.842 | 0.950 | 0.956 |
| iWildCam-OOD | 0.848 | 0.870 | 0.868 |
| Camelyon17-OOD | 0.936 | 0.945 | 0.949 |
| DomainNet-OOD | 0.955 | 0.977 | 0.968 |

Table 6: Results of Confusion matrix. Rank correlation ($\rho$) is the metric.

| ImageNet-R | 1% | 5% | 10% | 30% | 100% |
|---|---|---|---|---|---|
| MaxPred | 0.618 | 0.623 | 0.612 | 0.615 | 0.613 |
| SoftGap | 0.798 | 0.803 | 0.794 | 0.799 | 0.797 |
| Entropy | 0.302 | 0.305 | 0.295 | 0.296 | 0.295 |
| InfoMax | 0.407 | 0.431 | 0.426 | 0.426 | 0.423 |
| SoftmaxCorr | 0.873 | 0.928 | 0.925 | 0.926 | 0.925 |

Table 7: Sensitivity analysis of all methods on test set sizes. We report $r$ and find that SoftmaxCorr is relatively stable.

| ImageNet-Blur | 1% | 5% | 10% | 30% | 100% |
|---|---|---|---|---|---|
| MaxPred | 0.863 | 0.877 | 0.878 | 0.878 | 0.877 |
| SoftGap | 0.883 | 0.895 | 0.897 | 0.898 | 0.898 |
| Entropy | 0.767 | 0.778 | 0.779 | 0.780 | 0.780 |
| InfoMax | 0.830 | 0.848 | 0.850 | 0.851 | 0.850 |
| SoftmaxCorr | 0.948 | 0.966 | 0.967 | 0.967 | 0.967 |

Table 8: Sensitivity analysis of all methods on test set sizes. We report $r$ and find that SoftmaxCorr is relatively stable.

| CINIC | 1% | 5% | 10% | 30% | 100% |
|---|---|---|---|---|---|
| MaxPred | 0.764 | 0.750 | 0.755 | 0.747 | 0.753 |
| SoftGap | 0.780 | 0.772 | 0.781 | 0.772 | 0.777 |
| Entropy | 0.749 | 0.733 | 0.733 | 0.717 | 0.722 |
| InfoMax | 0.753 | 0.740 | 0.745 | 0.732 | 0.735 |
| SoftmaxCorr | 0.655 | 0.777 | 0.827 | 0.857 | 0.844 |

Table 9: Sensitivity analysis of all methods on test set sizes. We report $r$ and find that SoftmaxCorr is relatively stable.

| iWildCam-OOD | 1% | 5% | 10% | 30% | 100% |
|---|---|---|---|---|---|
| MaxPred | 0.820 | 0.873 | 0.893 | 0.906 | 0.917 |
| SoftGap | 0.850 | 0.902 | 0.918 | 0.926 | 0.936 |
| Entropy | 0.734 | 0.793 | 0.821 | 0.839 | 0.853 |
| InfoMax | 0.784 | 0.832 | 0.859 | 0.869 | 0.883 |
| SoftmaxCorr | 0.826 | 0.874 | 0.896 | 0.903 | 0.916 |

Table 10: Sensitivity analysis of all methods on test set sizes. We report $r$ and find that Softmax-Corr is relatively stable.

In Tables 7, 8, 9 and 10, we observe that 1) when test size is very small ($\geq 5\%$), all methods have slightly lower correlation strength. For example, on ImageNet-R, InfoMax and SoftmaxCorr drop 0.024 and 0.055, respectively; 2) all methods have stable correlation results when test data is moderately sufficient. Specifically, when the correlation of all methods becomes robust when the sampled ratio is larger than $10\%$.

## A.8 RESULTS OF SOFTMAX CORRELATION

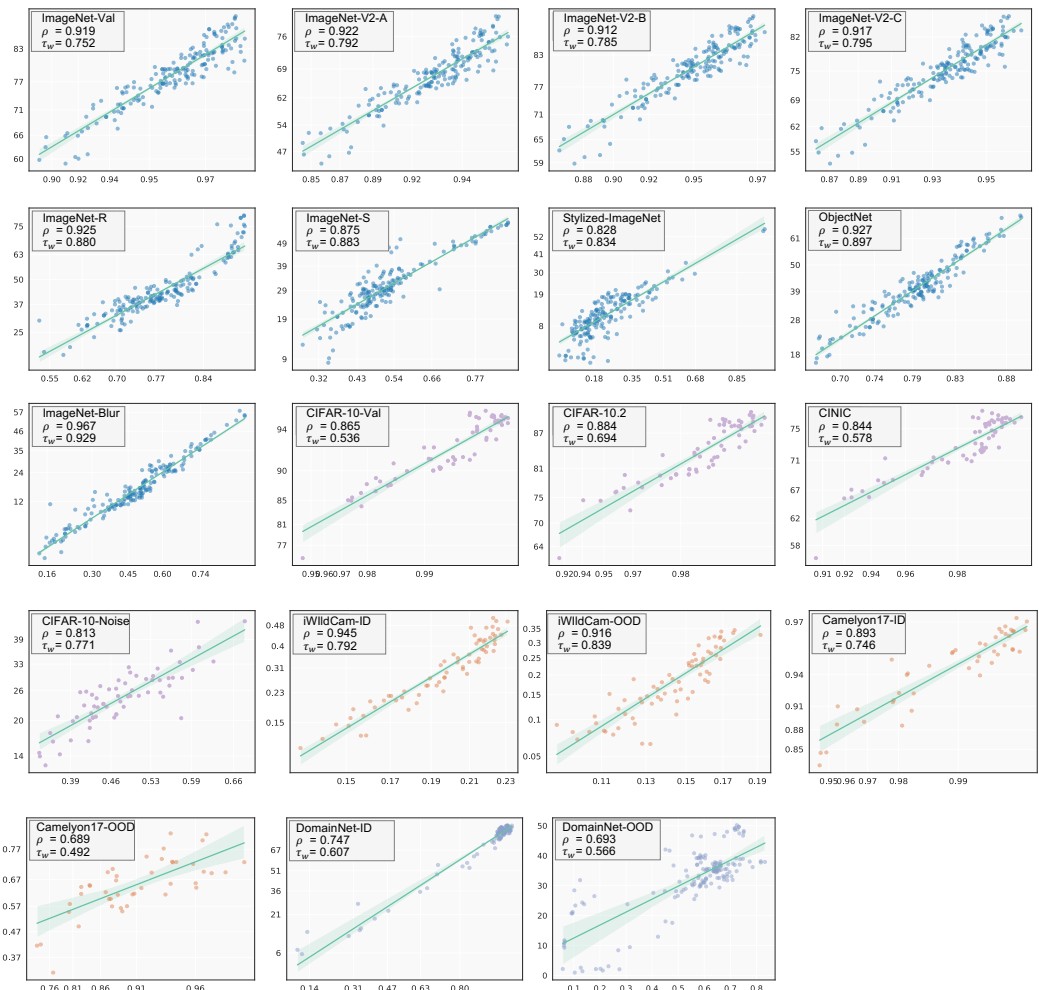

Figure 6: SoftmaxCorr *vs.* model generalization under ImageNet, CIFAR-10 and WILDS setups. The $y$-axis is accuracy, accuracy and macro-F1 for three setups, respectively. The straight lines are fit with robust linear regression (Huber, 2011). Axes are probit scaled as described in Section 3.

## A.9 RESULTS OF MAXIMUM PREDICTION PROBABILITY

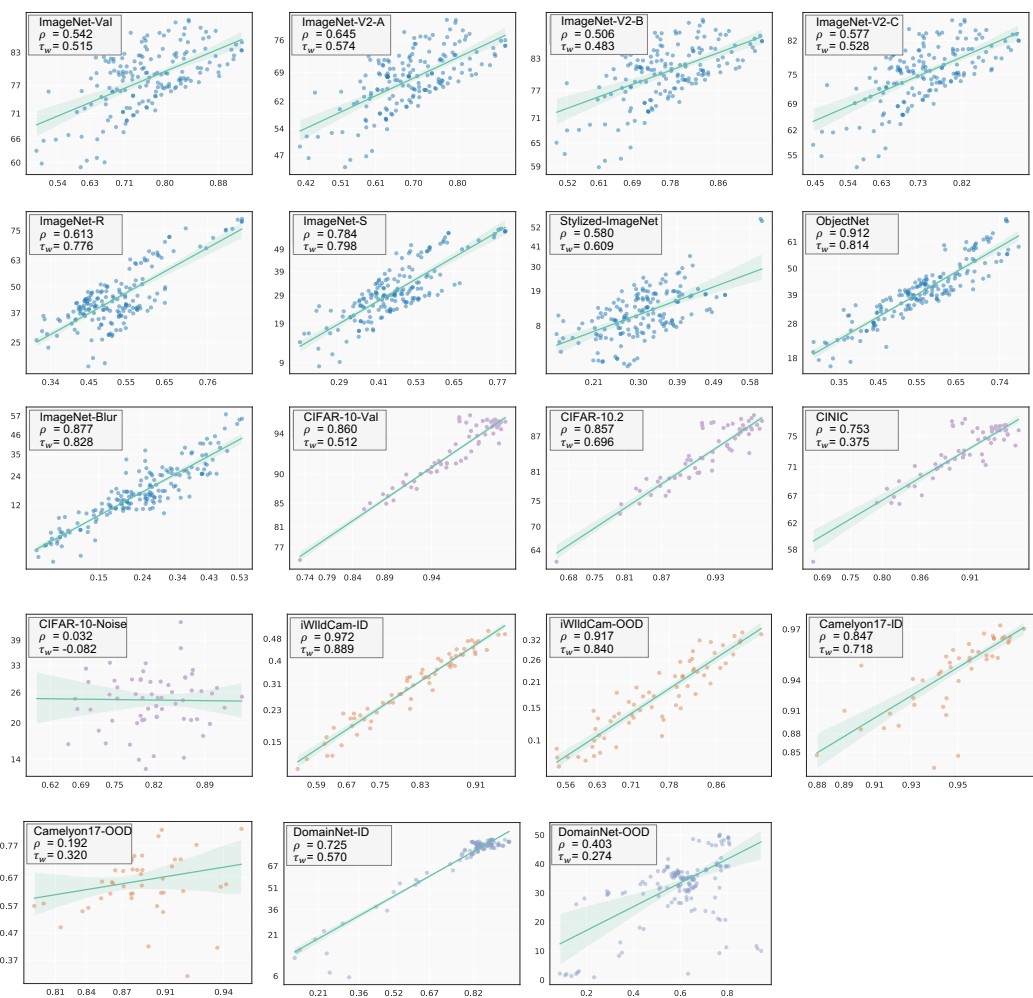

Figure 7: MaxPred *vs.* model generalization under ImageNet, CIFAR-10 and WILDS setups. The *y*-axis is accuracy, accuracy and macro-F1 for three setups, respectively. The straight lines are fit with robust linear regression (Huber, 2011). Axes are probit scaled as described in Section 3.

## A.10 RESULTS OF SOFTMAX GAP

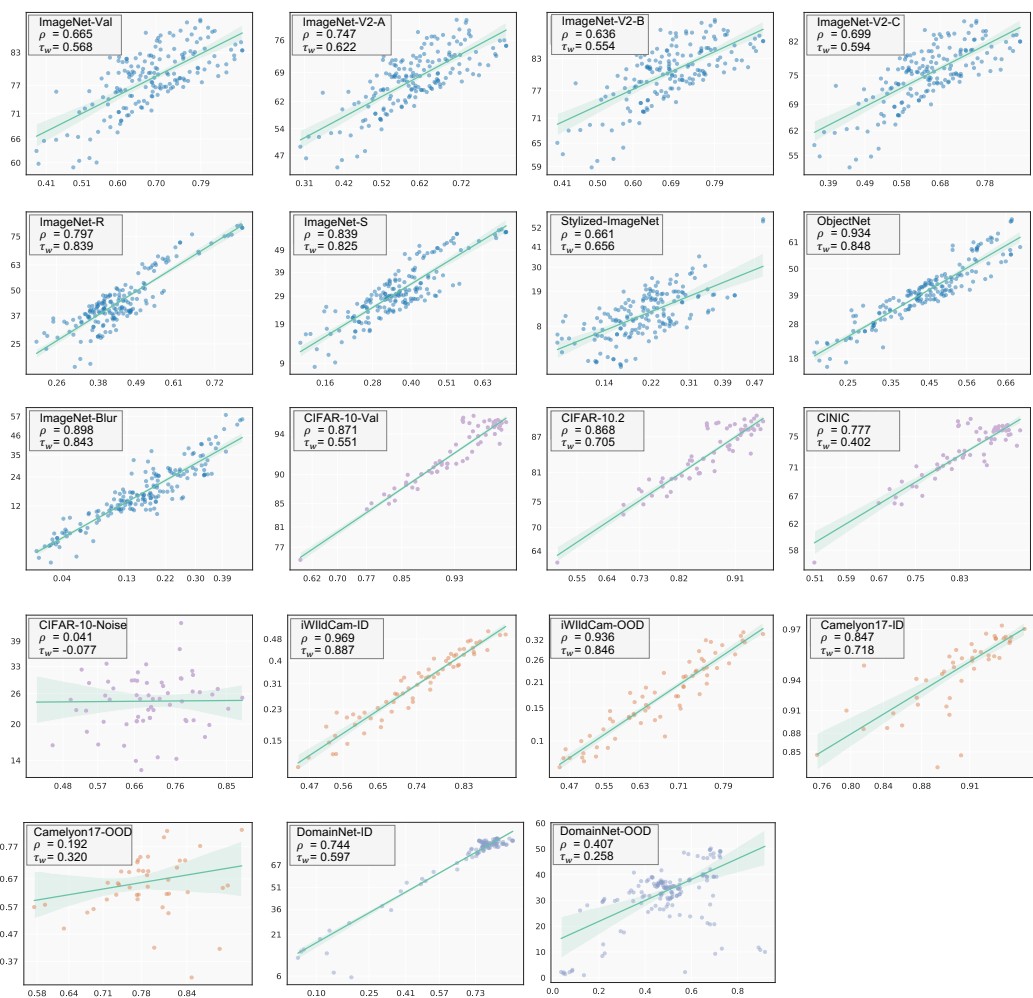

Figure 8: SoftGap *vs.* model generalization under ImageNet, CIFAR-10 and WILDS setups. The *y*-axis is accuracy, accuracy and macro-F1 for three setups, respectively. The straight lines are fit with robust linear regression (Huber, 2011). Axes are probit scaled as described in Section 3.

## A.11 RESULTS OF NEGATIVE PREDICTION ENTROPY

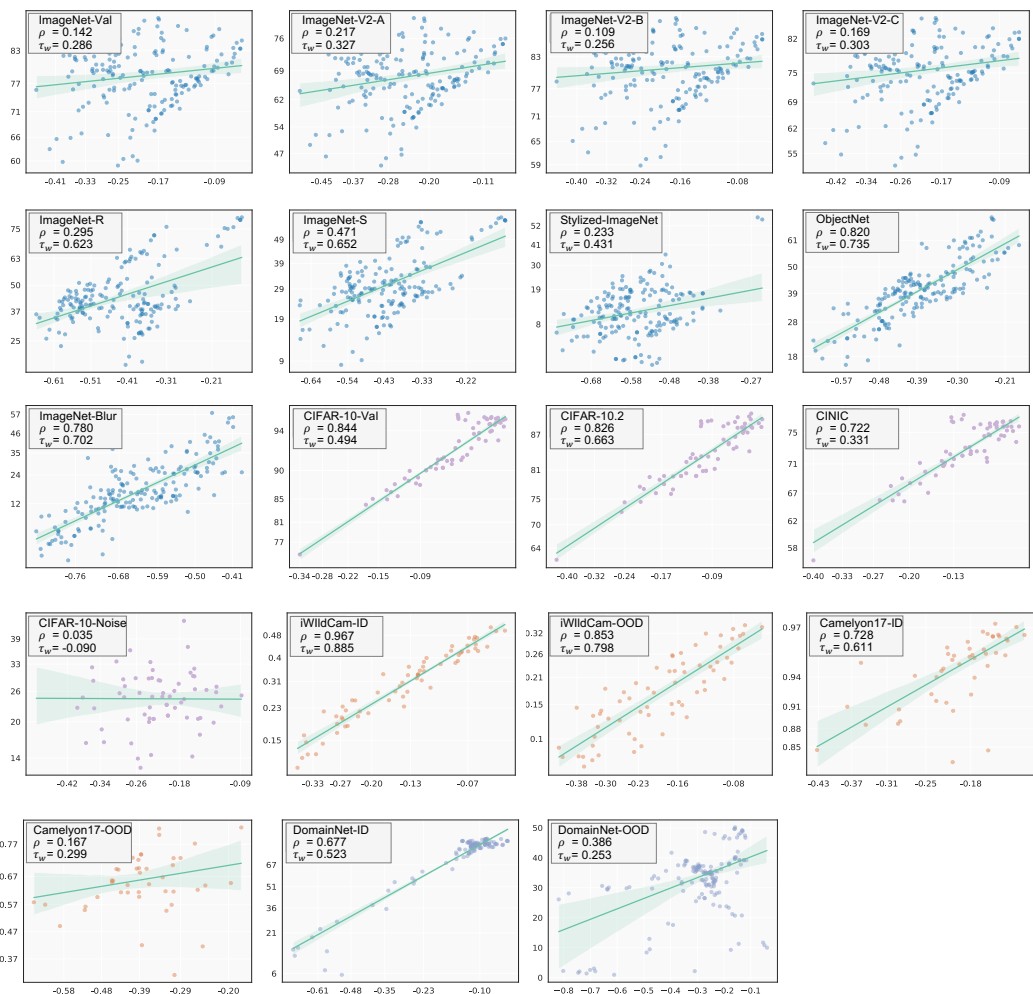

Figure 9: Entropy *vs.* model generalization under ImageNet, CIFAR-10 and WILDS setups. The *y*-axis is accuracy, accuracy and macro-F1 for three setups, respectively. The straight lines are fit with robust linear regression (Huber, 2011). Axes are probit scaled as described in Section 3.

## A.12 RESULTS OF INFORMATION MAXIMIZATION

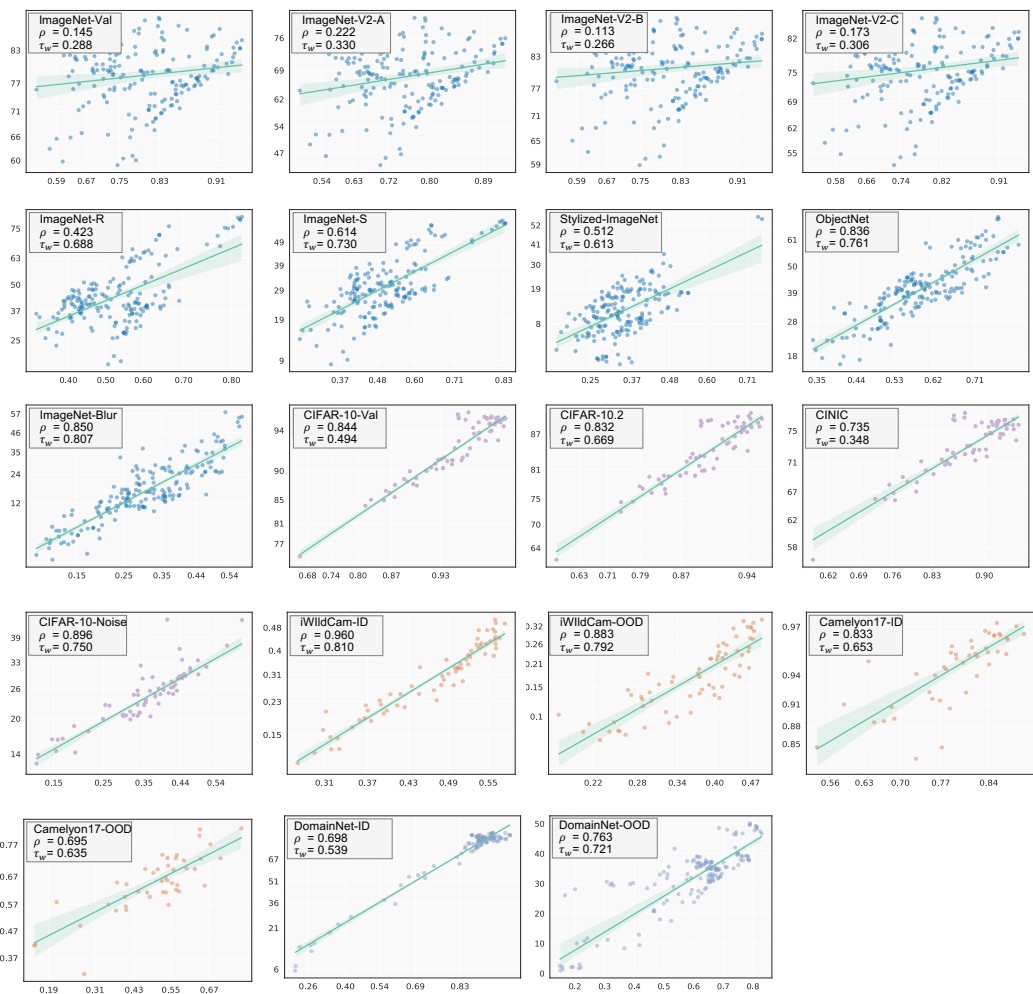

Figure 10: InfoMax *vs.* model generalization under ImageNet, CIFAR-10 and WILDS setups. The *y*-axis is accuracy, accuracy and macro-F1 for three setups, respectively. The straight lines are fit with robust linear regression (Huber, 2011). Axes are probit scaled as described in Section 3.

