# OpenReview forum: "Assessing Model Out-of-distribution Generalization with Softmax Prediction Probability Baselines and A Correlation Method"
_ICLR.cc/2023/Conference — Submitted to ICLR 2023_

### Official Review · Reviewer_Ku6q · 2022-10-25

**Confidence:** 4
**Correctness:** 4
**Technical Novelty And Significance:** 3
**Empirical Novelty And Significance:** 3
**Recommendation:** 6

**Clarity, Quality, Novelty And Reproducibility:**

Quality
---

Overall, the paper is of high quality, and extensive experimental results are presented. One experiment that is missing is an analysis of how sample size factors into the method's performance. Currently, it is unclear if a large test set is required for good performance. A stratified analysis of all methods under various test set sizes would help to answer this question.

Clarity
---

The paper is also relatively well written, though the description of the proposed method itself is woefully short. I find the paragraph starting with "Our motivation is two-fold..." difficult to follow, and expanding on these points would be useful. In my opinion, this would be worth moving some experiments to the appendix, as the core contribution of the work is this method, and currently the paper lacks details surrounding its motivation and explanation.

**Strength And Weaknesses:**

Strengths
---

- The paper makes experimental comparisons across a wide range of problems and models.
- The paper is reasonably well written and easy to follow.
- To the best of my knowledge, the proposed method is novel and addresses an important problem.

Weaknesses
---

- The proposed method is not always the best performing, and test sample size is not considered (see below).
- The paper would benefit from additional motivation and discussion around the method itself, which currently is explained only very briefly.

**Summary Of The Paper:**

This paper proposes an approach for estimating the out-of-distribution performance of a model using unlabeled test examples. The approach is relatively simple and involves computing the class correlation matrix based on the model's predictions on the text examples. Across a wide range of problems, models, and even training checkpoints, the proposed method shows strong empirical performance compared to well established baselines.

**Summary Of The Review:**

In summary, the paper is reasonably well written and the experiments are sound, and I have recommended a few additions that will further strengthen the paper. I am recommending weak accept and am happy to discuss further.

Edit after author response
---

Apologies for the late reply. I appreciate the authors' comments, clarifications, and revisions, and I am maintaining my recommendation of acceptance.

---

> ### Author Response · Authors · 2022-11-17
> **Author response to Reviewer Ku6q (Part II)**
>
> **Q3: One experiment that is missing is an analysis of how sample size factors into the method's performance. Currently, it is unclear if a large test set is required for good performance. A stratified analysis of all methods under various test set sizes would help to answer this question.**
>
> Thank you for your insightful suggestions. During the rebuttal, we conducted a stratified analysis of all five methods on four OOD datasets: ImageNet-R, ImageNet-Blur, CINIC and iWildCam-OOD.
>
> Specifically, we test different sizes of OOD datasets by randomly sampling 1%, 5%, 10%, 30% of test data. For each method under each test set size, we report the Spearman's rank correlation ($\rho$) of the average results of three random runs.
>
> In the following Table, we observe that ***1)*** when the test size is very small ($\le$ 5\%), all methods have slightly lower correlation strength. For example, on ImageNet-R, InfoMax and SoftmaxCorr drop 0.024 and 0.055, respectively; ***2)*** all methods have stable correlation results when test data is moderately sufficient. Specifically, ***when the correlation of all methods becomes robust when the sampled ratio is larger than 10%***.
>
> (1) Analysis on ImageNet-R:
> |ImageNet-R| 1% | 5% | 10% | 30% | 100% |
> | :---: | :---: | :---: | :---: | :---: | :---: |
> | MaxPred |0.618|0.623|0.612|0.615|0.613|
> | SoftGap |0.798|0.803|0.794|0.799|0.797|
> |Entropy|0.302|0.305|0.295|0.296|0.295|
> |InfoMax|0.407|0.431|0.426|0.426|0.423|
> |SoftmaxCorr|0.873|0.928|0.925|0.926|0.925|
>
> (2) Analysis on ImageNet-Blur:
> |ImageNet-Blur| 1% | 5% | 10% | 30% | 100% |
> | :---: | :---: | :---: | :---: | :---: | :---: |
> | MaxPred |0.863|0.877|0.878|0.878|0.877|
> | SoftGap |0.883|0.895|0.897|0.898|0.898|
> |Entropy|0.767|0.778|0.779|0.780|0.780|
> |InfoMax|0.830|0.848|0.850|0.851|0.850|
> |SoftmaxCorr|0.948|0.966|0.967|0.967|0.967|
>
> (3) Analysis on CINIC:
> |CINIC| 1% | 5% | 10% | 30% | 100% |
> | :---: | :---: | :---: | :---: | :---: | :---: |
> | MaxPred |0.764|0.750|0.755|0.747|0.753|
> | SoftGap |0.780|0.772|0.781|0.772|0.777|
> |Entropy|0.749|0.733|0.733|0.717|0.722|
> |InfoMax|0.753|0.740|0.745|0.732|0.735|
> |SoftmaxCorr|0.655|0.777|0.827|0.857|0.844|
>
> (4) Analysis on iWildCam-OOD:
> |iWildCam-OOD| 1% | 5% | 10% | 30% | 100% |
> | :---: | :---: | :---: | :---: | :---: | :---: |
> | MaxPred |0.820|0.873|0.893|0.906|0.917|
> | SoftGap |0.850|0.902|0.918|0.926|0.936|
> |Entropy|0.734|0.793|0.821|0.839|0.853|
> |InfoMax|0.784|0.832|0.859|0.869|0.883|
> |SoftmaxCorr|0.826|0.874|0.896|0.903|0.916|
>
> We have also included the above analysis in the revised paper (Section 5.4).

---

> ### Author Response · Authors · 2022-11-17
> **Author response to Reviewer Ku6q (Part I)**
>
> **Q1: The paper would benefit from additional motivation and discussion around the method itself, which currently is explained only very briefly. I find the paragraph starting with "Our motivation is two-fold..." difficult to follow, and expanding on these points would be useful.**
>
> Thank you for your valuable advice. We have revised Section 4.3 to further clarify our motivation and explain the method in more details. Specifically, we use the following sentences:
>
>    *Based on class correlation matrix $C$, we develop SoftmaxCorr to take into account the two characteristics. First, whether the model produces confident predictions and thus its computed class correlation matrix $C$ has a high intra-class correlation. Second, whether the model gives diverse predictions where all classes are predicted. This detects the trivial model solution where all data are confidently predicted as the same class. To achieve this, we define SoftmaxCorr as the cosine similarity between the class-class correlation matrix $C$ and an identity matrix $I_K$: $cos(C, I_K) = \frac{vec(C) \cdot vec(I_K)}{\lVert vec(C) \rVert  \lVert vec(I_K) \rVert}$, where $vec(\cdot)$ is the vectorization that converts the matrix into a column vector. $\lVert vec(C) \rVert$ and $\lVert vec(I_K) \rVert$ means the $L2$ norm of vectorized matrices $C$ and $I_K$.*
>
>   *A higher similarity score means that model gives 1) high prediction certainty (intra-class correlation) and 2) high prediction diversity (the diagonal elements of class correlation matrix is uniformly distributed).*
>
>
> Furthermore, we ***conducted more detailed analysis*** by studying the diagonal of the class-class correlation matrix. Specifically, we test the negative standard deviation of the diagonal elements in the class correlation matrix (***Diag-std***). We also report the sum of the diagonal entries of the class correlation matrix (***Diag-sum***).  In the following Table, we compare SoftmaxCorr with the two variants and report Spearman’s rank correlation ($\rho$).
>
> |Dataset|Diag-sum|Diag-std|SoftmaxCorr|
> | :---: | :---: | :---: | :---: |
> | ImageNet-V2-A |0.503|0.099|0.922|
> | ImageNet-R |0.669|0.314|0.925|
> |CIFAR-10.2|0.854|0.688|0.884|
> |CINIC|0.752|0.781|0.844|
> |iWildCam-OOD|0.916|-0.696|0.916|
> |Camelyon17-OOD|0.174|0.581|0.689|
> |Average|0.645|0.294|0.863|
>
> We observe that SoftmaxCorr is more predictive of OOD performance compared to Diag-sum and Diag-std on six datasets. *This suggests it is important to measure both prediction diversity and certainty for OOD generalization assessment*.
>
> **Q2: The proposed method is not always the best performing.**
>
> We would like to clarify that ***SoftmaxCorr generally is more indicative of model generalization ability than other measures***. It is true that SoftmaxCorr gives lower Spearman's rank correlations than a few other measures in some OOD test sets. However, we would like to point out that these measures lead to very unstable correlations under different setups. For example, while InfoMax presents a strong correlation ($\rho = 0.896$) on CIFAR-10-Noise, it exhibits a much weaker correlation ($\rho = 0.222$) on ImageNet-V2-A. In comparison, SoftmaxCorr has overall strong correlations and relatively more stable performance.

---

> ### Author Response · Authors · 2022-11-27
> **Thank you for your updated comment**
>
> Dear Reviewer Ku6q,
>
> Thank you for your positive assessment and valuable suggestions. We gratefully acknowledge that your comments helped us to improve our paper.
>
> Best,
>
> Authors

---

### Official Review · Reviewer_hesX · 2022-10-25

**Confidence:** 4
**Correctness:** 3
**Technical Novelty And Significance:** 3
**Empirical Novelty And Significance:** 3
**Recommendation:** 5

**Clarity, Quality, Novelty And Reproducibility:**

Clarity: mostly clear.
Reproducibility: code not released.


**Strength And Weaknesses:**

Strengths.
1. Developing predictive measures for accuracy is important and has several practical applications from ML safety to adaptation.
2. Their method is simple and computationally inexpensive to derive.


Weakness/questions.
1. Cosine-similarity of matrices is not formally defined and there is no standard definition of the same either.
2. Infomax vs SoftmaxCorr. Both the methods of estimation use almost similar information. They both look at diversity and confidence of predictions, but why is SoftmaxCorr so much better than InfoMax?
3. Variant B and SoftmaxCorr of Table 2. They are very similar, yet on wilds dataset, variant-b is far worse (even negative). Variant-B, however, considers only the diagonal elements and I expect it should be even closer to the Identity matrix when compared with SoftmaxCorr.
4. In all the figures, the authors should clarify what different data points (dots) in each plot are. Are these evaluations from different models? If so, how are they obtained and how do they differ?
5. Lack of understanding. In the discussion section, authors remark about how in-domain validation is not always informative of out-domain and that SoftmaxCorr could be more universal. But I do not see what merits SoftmaxCorr to be universal, on one of the WILDS dataset, we see softmaxCorr does not predict accuracy well. I would like to see the authors comment on when and why SoftmaxCorr is expected to work.
6. More evaluation. There have been several empirically derived measures of accuracy prediction, but the authors only compared with simple approaches that exploit strictly lower information from softmax probabilities (such as max probability or difference between highest two probabilities). Authors should compare with the most recent related work on this problem. The case of low performance on Camelyon-OOD should be further investigated and explained.
7. The motivation of Section 5.5 is unclear, especially in the last sentence.



**Summary Of The Paper:**

This work studies qualitative measures using prediction probabilities that can predict the out-of-domain generalization. They argue that softmax probabilities are informative of accuracy in new domains and moreover measures that maximally exploit structure in prediction probability are at an advantage.

**Summary Of The Review:**

The paper is mostly well-written and does a good job with its thorough systematic experiments. But some of the results and their method needs more explanation as further explained by my questions.

--------------
**Response to author comments**
I thank the authors for providing detailed clarification. Some of my concerns related to correctness of their results (such as questions 3, 4) are now addressed. I have also read other reviews and author responses. Reviewer 8Aud and I share similar concerns regarding comparison with other baselines, which the authors added in response. Accuracy-on-the-line explains accuracy poorly on 3/14 datasets while SoftmaxCorr is more consistent. However, it's unclear why the SoftmaxCorr is more consistent, even intuitively. Also, the definition of SoftmaxCorr is still not clear, what is $C.I$, elementwise product?

In summary, SoftmaxCorr is thoroughly evaluated and performs consistently well. Yet, I do not find the paper exciting. It is known that prediction probabilities are a good proxy for predicting accuracies and this work suggests that one could perform even better by also considering diversity of predictions. SoftmaxCorr is motivated from symptoms of a poor classifier: uncertain and skewed classification. So, a random classifier could as well be expected to have good performance on any dataset, which is why it is important to characterize and explicitly state the assumptions on the classifier and OOD datasets (and its relation to training data). The author's response to when SoftmaxCorr works is answered with more numbers on why it is important to look at diversity of predictions (along with confidence), and hence does not address my question. Also, the response on why SoftmaxCorr suffers on Camelyon-OOD is unconvincing. They explain that it could be due to underspecification, but why is underspecification only seen on this dataset and why does SoftmaxCorr fail for *underspecifed* setting despite not using ID datasets for accuracy predictions?

This work does not contain any surprising findings that I would like to eagerly share with a broader audience. For this reason, I recommend rejection and retain my original score.

---

> ### Author Response · Authors · 2022-11-17
> **Author response to Reviewer hesX (Part III)**
>
> **Q6: Authors should compare with the most recent related work on this problem.**
>
> Good suggestion. We further include one non-softmax probability-based method: adversarial input margin (Complexity) which measures the model complexity (Baldock et al., 2021). It is computed by the smallest norm required for an adversarial perturbation in the input to change the model’s class prediction. Please refer to Q1-4 / Reviewer 8Aud for more details. In addition, we include two recent methods that are softmax probability-based:  MaxLogit (Hendrycks et al., 2022) and Energy (Liu et al., 2020). We conduct the correlation study on ImageNet-V2-A, ImageNet-S and CIFAR-10.2 and show the full results in the following Table:
>
> (1) Method comparison under $\rho$
> |$\rho$|MaxLogit|Energy|Complexity|SoftmaxCorr|
> | :---: | :---: | :---: | :---: | :---: |
> |ImageNet-V2-A|-0.267|0.350|0.341|0.922|
> |ImageNet-S|0.024|0.087|0.329|0.875|
> | CIFAR-10.2 |0.210|-0.180|0.680|0.884|
>
>
> (2) Method comparison under $\tau_w$
>
> |$\tau_w$|MaxLogit|Energy|Model Complexity|SoftmaxCorr|
> | :---: | :---: | :---: | :---: | :---: |
> |ImageNet-V2-A|-0.237|0.158|0.282|0.792|
> |ImageNet-S|0.422|-0.180|0.417|0.883|
> |CIFAR-10.2|-0.032|-0.183|0.524|0.694|
>
> According to the results, we observe that both MaxLogit and Energy cannot exhibit good correlation. While model complexity archives some correlation on three datasets, it is still inferior to SoftmaxCorr.
>
> We have discussed the three new baselines in the revised paper (Section A.4).
>
>
> Hendrycks et al., Scaling out-of-distribution detection for real-world settings. In ICML, 2022.
>
> Baldock et al. Deep learning through the lens of example difficulty. In NeurIPS, 2021.
>
> Liu et al. Energy-based out-of-distribution detection. In NeurIPS, 2020.
>
> **Q7: The case of low performance on Camelyon-OOD should be further investigated and explained.**
>
> Thanks for your valuable advice. Camelyon17 is a binary image classification task where each example is a tissue patch. Its images are metastasized breast cancer tissue samples collected from different hospitals. As noted by Koh et al.,(2021), the held-out OOD test set contains tissue samples from a hospital not seen in the training set. ***The distribution shift largely arises from differences in staining and imaging protocols across hospitals***.
>
> In our study, we observe a similar ***underspecification phenomenon*** as (Miller et al., 2021; Wenzel et al., 2022): very similar ID performances lead to very different OOD performances. This indicates that the models cannot learn robust patterns that can be transferred to OOD data. Moreover, ***the similarity between images*** from the same slide or hospital is very high (Miller et al., 2021). This probably exacerbates the underspecification. Based on the above, we speculate that Softmaxcorr is less effective in detecting the underspecification issue on Camelyon-OOD and thus has low correlation. That said, it would be interesting to further study this phenomenon in future research.
>
> We have included the above discussion in the revised version (Section A.3).
>
> *Koh et al., Wilds: A benchmark of in-the-wild distribution shifts. In ICML, 2021*
>
> *Miller et al. Accuracy on the line: on the strong correlation between out-of-distribution and in-distribution generalization. In ICML, 2021*
>
> *Wenzel et al. Assaying out-of-distribution generalization in transfer learning. In  NeurIPS, 2022*
>
> **Q8: The motivation of Section 5.5 (DomainNet setup) is unclear, especially in the last sentence.**
>
> Thanks for pointing out this. On ImageNet, CIFAR and WILDS setups, all models are trained by standard empirical risk minimization and do not use the unlabeled OOD samples  for training. In some scenarios, domain adaptation (DA) algorithms are employed for learning target-adaptive models with additional unlabeled OOD samples.
>
> To explore whether SoftmaxCorr is still effective to assess the generalization of these models, we conduct correlation study under DomainNet setup. The models are trained by 9 different DA algorithms (e.g., DeepCORAL and DANN).
>
> In Table 1, we observe that SoftmaxCorr performs reasonably on DomainNet-ID and DomainNet-OOD. We also notice that InfoMax is better than SoftmaxCorr on DomainNet-OOD. We speculate that InfoMax is commonly used as a regularization in DA and it might better reflect models’ target-adaptation ability and exhibit higher correlation on DomainNet-OOD.
>
> We have updated Section 5.6 and clarified our motivation.

---

> ### Author Response · Authors · 2022-11-17
> **Author response to Reviewer hesX (Part II)**
>
> **Q5-1: Authors remark about how in-domain validation is not always informative of out-domain and that SoftmaxCorr could be more universal. It is unclear what merits SoftmaxCorr to be universal, on one of the WILDS dataset, it can be noticed that softmaxCorr does not predict accuracy well.**
>
> Thanks for this comment. We would like to clarify that we do not claim SoftmaxCorr will perform well under all circumstances, but precisely it is overall a good choice when labeled ID data is not accessible or accuracy-on-the-line does not hold.
>
> Specifically, we compare SoftmaxCorr with the accuracy-on-the-line under various datasets (please refer to ***Q1-1/ Reviewer 8Aud***). We observe that accuracy-on-the-line exhibits very weak correlation on CIFAR-10-Noise, Camelyon17-OOD and DomainNet-OOD. This is also pointed out by its authors. In comparison, SoftmaxCorr remains relatively informative and exhibits moderately high correlation on the three test sets. Moreover, when accuracy-on-the-line shows very strong correlations, SoftmaxCorr is also competitive. Moreover, we observe that Softmaxcorr is generally more stable than other measures across various datasets under three different setups
>
> **Q5-2: I would like to see the authors comment on when and why SoftmaxCorr is expected to work.**
>
> Thanks for this valuable suggestion.
>
> [Discussion on why SoftmaxCorr works] SoftmaxCorr uses class-wise relationships encoded by the softmax prediction probabilities. It considers both certainty and diversity of model predictions. First, when a model makes certain predictions, it is expected to see a large sum of diagonal elements in the computed class correlation matrix. Second, for the diverse predictions, the class correlation matrix is expected to have uniformaly distributed values on each diagonal entry. To simultaneously consider them, SoftmaxCorr uses the cosine similarity between the class-class correlation matrix and an identity matrix.
>
> To further explore SoftmaxCorr, we conducted more detailed analysis by studying the diagonal of class-class correlation matrix. Specifically, we test the negative standard deviation of the diagonal elements in the class correlation matrix (Diag-std). We also report the sum of the diagonal entries of the class correlation matrix (Diag-sum).  In the following Table, we compare SoftmaxCorr with the two variants and report Spearman’s rank correlation ($\rho$).
>
> |Dataset|Diag-sum|Diag-std|SoftmaxCorr|
> | :---: | :---: | :---: | :---: |
> | ImageNet-V2-A |0.503|0.099|0.922|
> | ImageNet-R |0.669|0.314|0.925|
> |CIFAR-10.2|0.854|0.688|0.884|
> |CINIC|0.752|0.781|0.844|
> |iWildCam-OOD|0.916|-0.696|0.916|
> |Camelyon17-OOD|0.174|0.581|0.689|
> |Average|0.645|0.294|0.863|
>
> We observe that SoftmaxCorr is more predictive of OOD performance compared to Diag-sum and Diag-std on six datasets. This suggests it is important to measure both prediction diversity and certainty for OOD generalization assessment.
>
> [Discussion on when SoftmaxCorr works] We consistently observe that Softmaxcorr exhibits a strong or moderately strong correlation with model generalization across different datasets and setups in Table 1. Based on this, we believe Softmaxcorr is practical and can work on most of the common scenarios. However, as we discussed in Section 6, Softmaxcorr implicitly assumes that the test set is balanced. While it works on some moderately imbalanced datasets (e.g., ImageNet-R and  ObjectNet), we think it might not be suitable for extremely imbalanced datasets. We hope to further understand this potential limitation and application score in future research.

---

> ### Author Response · Authors · 2022-11-17
> **Author response to Reviewer hesX (Part I)**
>
> **Q1: Cosine-similarity between matrices is not formally defined and there is no standard definition either.**
>
> Thanks for pointing out this. We define SoftmaxCorr as the cosine similarity between the class-class correlation matrix $C$ and an identity matrix $I_K$: $cos(C, I_K) = \frac{vec(C) \cdot vec(I_K)}{\lVert vec(C) \rVert  \lVert vec(I_K) \rVert}$, where $vec(\cdot)$ is the vectorization that converts the matrix into a column vector. $\lVert vec(C) \rVert$ and $\lVert vec(I_K) \rVert$ means the $L2$ norm of vectorized matrices $C$ and $I_K$. We have illustrated it in the revised paper (Section 4.3).
>
> **Q2: InfoMax vs SoftmaxCorr use almost similar information. Why is SoftmaxCorr so much better than InfoMax?**
>
> Good question. InfoMax and SoftmaxCorr consider prediction certainty and diversity in different manners. SoftmaxCorr exploits the class confusion information encoded in  the class correlation matrix. In comparison, InfoMax is an information-theoretical measure which considers the diversity by the entropy of empirical class distribution (first term) and the uncertainty by the average entropy of test samples (second term).
>
> In our study, we empirically observe that the average entropy of the test sample (Entropy in Table 1) could not well capture the prediction uncertainty on some datasets (e.g., ImageNet-R and ImageNet-S) and thus exhibits weak correlation. In contrast, the manner of SoftmaxCorr is shown to be effective and robust on different datasets. This might explain why InfoMax exhibits a weaker correlation strength than SoftmaxCorr.
>
> Based on the above, we think it would be promising to explore other ways to characterize both prediction uncertainty and diversity.
>
> **Q3: Variant B and SoftmaxCorr are very similar. Yet, on wilds dataset, variant-B is far worse. Variant-B is expected to be closer to the Identity matrix when compared with SoftmaxCorr.**
>
> Thank you for your meticulous review. Based on the cosine similarity calculation (please see Q1 above),  Variant-B and SoftmaxCorr are indeed very similar and Variant-B is closer to the identity matrix. After carefully checking the results, we corrected that the correlation on iWildCam-OOD is positive. Please see the following Table:
>
> |Dataset| Variant-A | Variant-B | SoftmaxCorr |
> | :---: | :---: | :---: | :---: |
> | ImageNet-V2-A |0.503|0.902|0.922|
> | ImageNet-R |0.669|0.924|0.925|
> |CIFAR-10.2|0.854|0.860|0.884|
> |CINIC|0.752|0.817|0.844|
> |iWildCam-OOD|0.916|0.914|0.916|
> |Camelyon17-OOD|0.174|0.597|0.689|
> |Average|0.645|0.836|0.863|
>
> **Q4: In all the figures, the authors should clarify what different data points (dots) in each plot are. Are these evaluations from different models? If so, how are they obtained and how do they differ?**
>
> Thank you for your valuable suggestions. In all figures, each point denotes a trained model. For different setups, models are trained for their corresponding tasks. Following the practice in (Miller et al., 2021; Taori et al., 2020), we include various models with diverse model architectures (e.g., ResNet, ViT and MLP-Mixer) and training schemes (e.g., data augmentations and learning schedulers).
>
> For each setup, the models are publicly available and we have introduced them in Section A.1 in Appendix. We have updated the captions for clarification.
>
> *Miller et al., Accuracy on the line: on the strong correlation between out-of-distribution and in-distribution generalization.  In ICML, 2021*
>
> *Taori et al., Measuring robustness to natural distribution shifts in image classification. In NeurIPS 2020*

---

> ### Author Response · Authors · 2022-11-27
> **Follow-Up Clarification (2/2)**
>
> > *"The response on why SoftmaxCorr suffers on Camelyon-OOD is unconvincing. They explain that it could be due to underspecification, but why is underspecification only seen on this dataset and why does SoftmaxCorr fail for underspecified settings despite not using ID datasets for accuracy predictions?"*
>
> Thanks for raising a further discussion on Camelyon17-OOD.
>
> - First, we would like to clarify that we did not say underspecification is only seen on Camelyon17-OOD. Instead, the underspecification is extreme on Camelyon17-OOD as also observed in (Miller et al., 2021; Wenzel et al., 2022): very similar ID performances lead to significantly different OOD performances.
>
>     The underspecification phenomenon depends on the datasets and models associated with the task (D’Amour et al. 2020). We think that the *models trained on Camelyon17-ID training set encode substantially different inductive biases* that result in significantly different OOD generalization behaviors. Furthermore, *the similarity between images* from the same slide or hospital is very high (Miller et al., 2021). This probably exacerbates the underspecification on Camelyon17-OOD.
>
>     *D’Amour, Alexander, et al. "Underspecification presents challenges for credibility in modern machine learning." Journal of Machine Learning Research, 2020*
>
> - Second, we also observe the underspecification on CIFAR-10 Noise (consistent with Miller et al., 2021), where the correlation achieved by SoftmaxCorr is moderately strong.
>
> - Third, it is true that the calculation of Softmaxcorr does not use ID datasets. Yet, the models are learned on Camelyon-17-ID training set and they encode substantially different inductive biases. The softmax probabilities they compute on Camelyon17-OOD could be affected by these inductive biases. This cannot be well captured by the prediction certainty and diversity only, so SoftmaxCorr exhibits relatively low correlation strength.
>
> Based on the above, we speculate that ***the prediction certainty and diversity used by Softmaxcorr may be insufficient in fully representing the underspecification*** (especially on Camelyon17-OOD) and thus has low correlation. That said, it would be interesting to study this phenomenon for improving the OOD measure.

---

> ### Author Response · Authors · 2022-11-27
> **Follow-Up Clarification (1/2)**
>
> Dear Reviewer hesX:
>
> We appreciate your detailed feedback and thank you for pointing out questions that need further clarification. Please see the clarification below:
>
> > *"Accuracy-on-the-line explains accuracy poorly on 3/14 datasets while SoftmaxCorr is more consistent. However, it's unclear why the SoftmaxCorr is more consistent, even intuitively”*
>
> *Accuracy-on-the-line does not consider the characteristics of the OOD dataset*: it only uses ID accuracy as the OOD measure. We observe that when OOD datasets exhibit special or unique characteristics, this method could be less effective. For example, CIFAR-10-Noise and CIFAR have different data co-variances, and the phenomenon of accuracy on the line does not hold. This is also pointed out by its authors.
>
> In comparison, *SoftmaxCorr is defined in terms of softmax-predicted probabilities computed on the OOD dataset*. That is, the information it considers is directly related to the OOD dataset (*i.e.*, prediction certainty and diversity on OOD samples). We show that both prediction certainty and diversity are informative and useful in assessing model generalization across various datasets with different types of distribution shifts.
>
> > *"The definition of SoftmaxCorr is still not clear."*
>
> Thank you for your meticulous comment. We define SoftmaxCorr as the cosine similarity between the class-class correlation matrix $C$ and an identity matrix $I_K$: $cos(C, I_K) = \frac{vec(C) \cdot vec(I_K)}{\lVert vec(C) \rVert  \lVert vec(I_K) \rVert}$, where $vec(\cdot)$ is the vectorization that converts the matrix into a column vector. $\lVert vec(C) \rVert$ and $\lVert vec(I_K) \rVert$ means the $L2$ norm of vectorized matrices $C$ and $I_K$.
>
> > *“... random classifier could as well be expected to have good performance on any dataset, which is why it is important to characterize and explicitly state the assumptions on the classifier and OOD datasets (and its relation to training data).”*
>
> Thanks for sharing this valuable comment. We agree that there could be some special cases where prediction uncertainty and diversity may be insufficient in assessing model generalization. Then, the assumptions on classifiers or OOD datasets are useful. Inspired by this valuable comment, we further ***discuss when SoftmaxCorr works by discussing its application scope***:
>
> - First, this work considers the model generalization ability and assumes the deep models are trained on the ID training set following (Miller et al., 2021; Taori et al., 2020). As suggested by the reviewer, if a random model can make both highly certain and highly diverse predictions, then SoftmaxCorr is less useful.
>
> - Second, when testing on various common datasets (Table 1), Softmaxcorr exhibits a strong or moderately strong correlation with model generalization. However, as we discussed in Section 6, while it works on some moderately imbalanced datasets (*e.g.*, ImageNet-R and  ObjectNet), we think it might not be suitable for extremely imbalanced datasets. Moreover, under very special conditions (*e.g.*, adversarial attack), SoftmaxCorr might be less effective in characterizing model generalization.

---

> ### Author Response · Authors · 2022-12-01
> **Clarification on Contribution**
>
> > *"In summary, SoftmaxCorr is thoroughly evaluated and performs consistently well. Yet, I do not find the paper exciting. It is known that prediction probabilities are a good proxy for predicting accuracies and this work suggests that one could perform even better by also considering the diversity of predictions."*
>
> > *"This work does not contain any surprising findings that I would like to eagerly share with a broader audience. For this reason, I recommend rejection and retain my original score"*
>
> We respectfully disagree. While the observed insights (both certainty and diversity are important) may not be surprising in hindsight, our work is the first to experimentally show that Softmax probability is useful for OOD generalization assessment. We further contribute a simple but effective measure SoftmaxCorr. By discussing how SoftmaxCorr works against alternatives/variants, we show others how to explore OOD generalization measures in future work.
>
> We would like to highlight our novelty and contribution from the following three aspects.
>
> - First, existing works study Softmax prediction probability for detecting open-set samples (Hendrycks and Gimpel, 2016) and estimating the accuracy of a single trained classifier (Guillory, et al., 2021; Garg, et al., 2022). In contrast, this work explores ***its new application in an important but underexplored task***: characterizing model OOD generalization, with the goal of reflecting and ranking the performance of various models on an OOD dataset.
>
> - Second, we propose a ***simple yet effective measure named SoftmaxCorr***, which uses class-wise relationships encoded by the softmax prediction probabilities. It considers both the certainty and diversity of model predictions. We thoroughly validate its effectiveness and show that it is preferable and more stable across different datasets compared with existing measures (e.g., adversarial input margin and entropy). We stress the fact that ***our simple method*** can be generalized to various datasets with different types of distribution shifts is a ***non-trivial discovery***.
>
> - Third, our ***new insight is that both prediction certainty and diversity are important and informative***. Specifically, our component analysis (Table 2) studies two variants/alternatives of SoftmaxCorr. We show that the two pieces of information are important and should be considered together.
>
> In summary, we believe ***our work is worth sharing with the community***.
>
> Thank you again for raising this further clarification. We hope our response has addressed the questions. We remain available and committed to incorporating further suggestions to improve our work.

---

### Official Review · Reviewer_8Aud · 2022-10-26

**Confidence:** 4
**Correctness:** 4
**Technical Novelty And Significance:** 2
**Empirical Novelty And Significance:** 2
**Recommendation:** 5

**Clarity, Quality, Novelty And Reproducibility:**

The paper is well written and clear. To my knowledge the SoftmaxCorr is novel. The paper makes use of open source models and code, which indicates ease of reproducibility.

**Strength And Weaknesses:**

**Strengths**

- The paper is well written and clear.
- SoftmaxCorr is easy to compute and the underlying idea is simple and intuitive.
- The comparison to the confusion matrix is compelling, Fig. 2.
- SoftmaxCorr outperforms other metrics computed solely from the softmax predictive probabilities.

**Weaknesses**

*Comparison to non-softmax probability baselines*

The motivation for restricting the baselines to metrics computed solely on the softmax predictive probabilities is not clear to me. It would be understandable if the authors compared to other methods which may have additional requirements such as architectural changes, availability of ID labels, etc. and then stated that such methods have disadvantages due to these extra requirements. But given that these requirements may not be unreasonably prohibitive it seems to me that it is necessary to add such baselines.

In particular:

- It would have been useful to compare against methods which make use of ID performance. W.r.t. "we emphasize that SoftmaxCorr is overall a good alternative when labeled ID data is inaccessible or accuracy-on-the-line does not hold", I regard it as atypical that labeled ID data would not be available so it would be reasonable to evaluate against such methods e.g. Miller et al. (2021) and/or Wenzel et al. (2022).
- SNGP, Liu et al. (2020) and DUQ, Van Amersfoort et al. (2020), are competing methods. While they require architectural changes, it would be important to contextualize the SoftmaxCorr results by comparing against these methods.
- A further interesting baseline, would be to use the confusion matrix directly (i.e. cosine similarity of the confusion matrix with the identity matrix). Clearly on new unlabelled data points we will not have labels with which to compute the confusion matrix. But given that the SoftmaxCorr metric must be computed on a full dataset rather than individual samples and that it effectively assumes a stationary OOD distribution, then it would not seem unreasonable to label a small number of these samples and then compute the confusion matrix directly. I am open to arguments as to why this may be impractical or undesirable in certain scenarios and naturally not requiring labels is an advantage of SoftmaxCorr in any case, nonetheless it would be useful to have this oracle and potentially practical method benchmarked in the experiments section.

*Evaluation on far-OOD and OOD datasets with labels spaces different to the ID dataset*

Identifying OOD samples which are very far from the ID dataset, so far as not belonging to the same label space, is a key application of OOD detection metrics. Methods such as SNGP and DUQ have the advantage of being applicable to such scenarios. One major concern I have with the paper is that the OOD datasets are not very far from the ID datasets, for example ImageNet variants for ImageNet ID dataset and CIFAR variants for the CIFAR-10 training dataset. In such cases ID test accuracy has been shown to be a good predictor of OOD generalization Minderer, et al (2021).

The SoftmaxCorr metric could be computed on OOD datasets that do not share a label space with ID dataset. It is possible that SoftmaxCorr by approximating the confusion matrix is simply a good proxy for ID test set accuracy or similar metrics and that these metrics and SoftmaxCorr will not generalize to far OOD datasets. I would have liked the authors to have stress tested their method by evaluating it on datasets such as OOD=SVHN when ID=CIFAR-10 as done in Liu et al. (2020) and Van Amersfoort et al. (2020), and similarly more challenging label shifted datasets for the ID=ImageNet experiments.

*Dataset level metric*

The SoftmaxCorr metric seems to be computed at the dataset level, this is quite a restrictive setting, meaning that individual samples cannot be scored and the metric cannot be directly applied to online prediction problems where inputs may come from a distribution shifting over time. I would be interested in the authors commenting on this disadvantage of the method. An interesting sensitivity analysis would be to progressively reduce the size of OOD dataset size N and measure the impact on the Table 1 results.




Liu, Jeremiah, et al. "Simple and principled uncertainty estimation with deterministic deep learning via distance awareness." Advances in Neural Information Processing Systems 33 (2020): 7498-7512.

Van Amersfoort, Joost, et al. "Uncertainty estimation using a single deep deterministic neural network." International conference on machine learning. PMLR, 2020.





**Summary Of The Paper:**

The authors propose a method, SoftmaxCorr, which is an easy to compute metric to evaluate the generalization performance of a neural network on an OOD dataset. The metric is computed as the cosine similarity between the identity matrix and a correlation matrix of the matrix of predictions on the OOD dataset. The metric empirically is shown to approximate well the confusion matrix. SoftmaxCorr compares favourably to other metrics which make use only of the network predictive probabilities to compute an OOD generalization metric, when evaluated on OOD datasets sharing the same label space as the ID dataset.

**Summary Of The Review:**

The paper is well written and presents a clear, simple and effective metric for predicting OOD generalization performance. SoftmaxCorr compares well to other methods which only make use of the OOD softmax probabilities. However I have two major concerns: (i) I would have liked to have seen evaluation against other practical baselines which do not only use the softmax probabilities and (ii) I would have liked to see evaluation against far-OOD datasets and OOD datasets with different label spaces to the ID dataset, as there is existing evidence that ID accuracy is a good predictor of OOD performance for the near OOD datasets used in the paper.

---

> ### Author Response · Authors · 2022-11-17
> **Author response to Reviewer 8Aud (Part IV)**
>
> **Q3-1: The SoftmaxCorr metric seems to be computed at the dataset level, this is quite a restrictive setting, meaning that individual samples cannot be scored and the metric cannot be directly applied to online prediction problems where inputs may come from a distribution shifting over time. I would be interested in the authors commenting on this disadvantage of the method.**
>
> Thanks for your valuable suggestion. We follow the standard setup in (Miller et al., 2021; Taori et al., 2020)  to study the OOD model generalization, where all OOD test data are given. For the online prediction problem, the test set size would be relatively small. That is, OOD data might be insufficient for OOD measures including SoftmaxCorr and potentially degrades their predictive ability. We have discussed this point in the revised paper.
>
> Baek et al., Agreement-on-the-line: Predicting the performance of neural networks under distribution shift. NeurIPS, 2022.
>
> Taori et al., Measuring robustness to natural distribution shifts in image classification. NeurIPS, 2020.
>
> **Q3-2: An interesting sensitivity analysis would be to progressively reduce the size of OOD dataset size N and measure the impact on the Table 1 results. An interesting sensitivity analysis would be to progressively reduce the size of OOD dataset size N and measure the impact on the Table 1 results.**
>
> Good suggestion. To study the robustness of SoftmaxCorr to test set size,  we conduct a sensitivity analysis by randomly sampling 1%, 5%, 10%, 30% of images on four OOD datasets (ImageNet-R, ImageNet-Blur, CINIC and iWildCam-OOD).
>
> In the following Table, we report the Spearman's rank correlation and the results are the average of three random runs.
>
> |Dataset| 1% | 5% | 10% | 30% | 100% |
> | :---: | :---: | :---: | :---: | :---: | :---: |
> ImageNet-V2-A|0.820|0.853|0.865|0.901|0.922
> |ImageNet-R|0.873|0.928|0.925|0.926|0.925|
> |CIFAR-Noise|0.734|0.785|0.814|0.813|0.813|
> |CINIC|0.655|0.777|0.827|0.857|0.844|
> |iWildCam-OOD|0.826|0.874|0.896|0.903|0.916|
> |Camelyon17-OOD | 0.708 | 0.701 | 0.690 | 0.694 | 0.689|
>
>
> We observe that when the number of test data is very small (1%), the correlation of SoftmaxCorr drops. When the dataset size increases ($\ge 5\%$), SoftmaxCorr gives high and stable correlation strength. This suggests that SoftmaxCorr requires a reasonable number of samples to capture model OOD generalization. Furthermore, we have a similar observation for other methods (please refer to Q3/ Reviewer Ku6q)
>
> We have included the above sensitivity analysis in the revised paper (Section 5.4).

---

> > ### Comment · Reviewer_8Aud · 2022-11-30
> > **Thank you for your response**
> >
> > Thanks to the authors for their constructive and useful response to my review. In particular I appreciate that the authors have now added the suggested accuracy-on-the-line and oracle confusion matrix baselines to the experimental results and have also conducted the suggested sensitivity analysis.
> >
> > ---
> >
> > "datasets whose accuracy strongly correlate with ID accuracy may not consequently be defined as near-ID"
> >
> > I agree this is the case and recognise that not all OOD datasets in the paper are near-OOD. However I did not wish to define near-OOD in this way. My point was that by more subjective measures SVHN is clearly further OOD from CIFAR-10 than CIFAR-10 variants with particular corruptions applied (similarly for ImageNet). Therefore I still see it as valid and important that the authors add more standard far-OOD datasets to stress test the method in a scenario in which I suspect, but I may be proven incorrect, may be particularly disadvantageous for the method.
> >
> > ---
> >
> > I am very borderline as to whether I will update my score. I really appreciate that the authors have given a detailed and comprehensive response. However I still have reservations about the method and problem setting. The problem setting, as clarified by the authors in the rebuttal, seems extremely restrictive (no access to any labels from the tested dataset, use of only the model softmax probabilities, general setup of model comparison on new dataset vs. testing a single model as to whether it is likely to generalize well on a given dataset). In addition it is not clear to me how necessary the SoftmaxCorr method is given that in most practical settings the confusion matrix approach, which has now been added by the authors, would be feasible to compute and performs very well. I will maintain my score for now, but I am happy to update it or be outvoted should the others reviewers reach a consensus for acceptance.

---

> > > ### Author Response · Authors · 2022-11-30
> > > **Follow-Up Clarification (3/3)**
> > >
> > > >*"It is not clear to me how necessary the SoftmaxCorr method is given that in most practical settings the confusion matrix approach, which has now been added by the authors, would be feasible to compute and performs very well."*
> > >
> > > We gratefully thank you for suggesting this oracle baseline. We think the strong correlation gained by the ***oracle does not compromise*** the contribution of SoftmaxCorr. It instead suggests that exploring class confusion information for OOD generalization assessment is feasible. Furthermore, we clarify that the oracle requires the labels, which ***could be impractical or undesirable*** as discussed in the above question: 1) it could be very expensive and challenging to label datasets that require expert-level knowledge. 2) It might not be feasible to label every dataset since the test sets can keep changing. In addition, we think it would be helpful and interesting to use labeled samples to further guide and improve SoftmaxCorr. Based on the above discussion, we think that Softmax is valid, necessary, and non-trivial.
> > >
> > > Thanks again for your follow-up comments, we hope the above clarification resolves the questions. Please let us know if you have any other comments. We remain available and eager to hear your feedback.
> > >
> > > Kind regards,
> > >
> > > Authors

---

> > > ### Author Response · Authors · 2022-11-30
> > > **Follow-Up Clarification (2/3)**
> > >
> > > >*"The problem setting, as clarified by the authors in the rebuttal, seems extremely restrictive (no access to any labels from the tested dataset, use of only the model softmax probabilities, general setup of model comparison on new dataset vs. testing a single model as to whether it is likely to generalize well on a given dataset)."*
> > >
> > > Thank you for this valuable suggestion. We clarify that the ***problem setting is not restrictive but reasonable and practical***. Please see the below discussion.
> > >
> > > - “no access to any labels from the tested dataset”. ***First***, the setting and our method do not assume a specific known OOD dataset. Instead, the testing distributions/datasets are various and often changing. In this case, a common assumption is the OOD dataset is unlabeled (Kouw and Loog, 2019) or unknown (Gulrajani & Lopez-Paz, 2021).
> > >
> > >     ***Second***, we agree with the reviewer that it would be possible to collect a small number of labeled data. However, we also highlight that ***this may be impractical or undesirable***: 1) it could be very expensive and challenging to label datasets that require expert-level knowledge. For example, precisely labeling wildlife categorization of iWildCam images and annotating cancer tissue images of Camelyon17. 2) it might not be feasible to label every test set. When testing distribution/environment changes, we need to label again.
> > >
> > >     *Kouw and Loog. A review of domain adaptation without target labels. TPAMI, 2019*
> > >
> > >     *Gulrajani and Lopez-Paz. In search of lost domain generalization. ICLR, 2021*
> > >
> > > - “use of only the model softmax probabilities”. We clarify that the ***Softmax probability is an efficient choice***. It is simple and very easy to compute. We ***further validate it is an effective choice*** by comparing it with the non-Softmax-based choice (adversarial input margin, Q1-4/ Reviewer 8Aud) and discussing the infeasibility of choices that require model changes and retraining (Q1-2/ Reviewer 8Aud). The extensive experiment shows that our softmaxCorr performs consistently well. Also, as discussed in Section 6, while this work mainly studies Softmax probability, we believe that studying the other information (*e.g.*, model weights) in future work would be interesting.
> > >
> > > - “general setup of model comparison on new dataset vs. testing a single model as to whether it is likely to generalize well on a given dataset”. This is an interesting idea. ***First***, our task setting extends the existing ID-generalization setting (Jiang et al., 2020) to the OOD case. The key focus of this work is exploring an OOD measure monotonically related to model generalization on a new dataset. This helps us characterize and analyze the models’ generalization behaviors.
> > >
> > >     ***Second***, testing how well a single model generalizes well on a given dataset is extensively studied in another task of unsupervised accuracy estimation (Deng and Zheng, 2021; Guillory et al., 2021; Garg et al., 2022). This task typically derives model-based distribution statistics of a test set. We note that these two tasks ***study the model generalization from two perspectives and they are complementary***.
> > >
> > >     ****Third****, we would like to highlight that our ***SoftmaxCorr is feasible to predict how well a single model generalizes*** on a new test set. Specifically, as shown in Fig. 5, we observe a strong correlation between SoftmaxCorr and model accuracy when testing a single model on various datasets. We will further emphasize this merit of SoftmaxCorr which will further enhance its contribution.
> > >
> > >     *Jiang et al. Neurips 2020 competition: Predicting generalization in deep learning. arXiv preprint arXiv:2012.07976, 2020*
> > >
> > >     *Deng and Zheng. Are labels always necessary for classifier accuracy evaluation? CVPR, 2021*
> > >
> > >     *Guillory et al. Predicting with confidence on unseen distributions. ICCV, 2021*
> > >
> > >     *Garg et al. Leveraging unlabeled data to predict out-of-distribution performance. ICLR, 2022*

---

> > > ### Author Response · Authors · 2022-11-30
> > > **Follow-Up Clarification (1/3)**
> > >
> > > Dear Reviewer 8Aud,
> > >
> > > We appreciate your thoughtful and constructive feedback! Thank you for further discussion on the far-OOD dataset, problem setting, and oracle confusion matrix baseline. We think these points further improve our work and highlight our contributions. Please see the clarifications below.
> > >
> > > > *...My point was that by more subjective measures SVHN is clearly further OOD from CIFAR-10 than CIFAR-10 variants with particular corruptions applied (similarly for ImageNet). Therefore I still see it as valid and important that the authors add more standard far-OOD datasets to stress test the method in a scenario in which I suspect, but I may be proven incorrect, may be particularly disadvantageous for the method."*
> > >
> > > Thank you for raising this constructive comment. We agree that SVHN is clearly far from CIFAR-10, because 1) their distributions are different and 2) their label spaces (or classes) are completely different. However, we would like to clarify that ***a dataset with a different label space is not suitable to measure model generalization***. Specifically, generalization refers to the model’s ability to classify the modeled or learned classes; the model cannot classify images from unseen classes.
> > >
> > > The suggested far-OOD case is where the task of out-of-distribution detection arises, with the goal of detecting and rejecting the images from unseen classes (Yang et al., 2022). In comparison, this work studies the OOD generalization (Miller et al., 2021; Wenzel et al., 2022), where the ID and OOD datasets share the same label space and the goal is to characterize the model performance under distribution shift.  We believe that ***the suggested far-OOD case is beyond the research scope of OOD generalization***. Therefore, we think the suggested far-OOD case ***might not be suitable*** to test the OOD measures including SoftmaxCorr.
> > >
> > > *Yang, et al. Openood: Benchmarking generalized out-of-distribution detection. NeurIPS, 2022*

---

> ### Author Response · Authors · 2022-11-17
> **Author response to Reviewer 8Aud (Part III)**
>
> **Q2-1: Evaluation on far-OOD: One major concern I have with the paper is that the OOD datasets are not very far from the ID datasets, for example ImageNet variants for ImageNet ID dataset.  In such cases ID test accuracy has been shown to be a good predictor of OOD generalization Minderer, et al (2021).**
>
> Thanks for raising this discussion. We would like to clarify that ***datasets whose accuracy strongly correlate with ID accuracy may not consequently be defined as near-ID***. For example, ID accuracy is predictive of OOD accuracy on ObjectNet. This does not mean ObjectNet is near-ID. Instead, it is a challenging far-ID dataset (with controls for rotation, background, and viewpoint), where most of state-of-the-art models exhibit 40-45% drop in performance.
>
> Moreover, we would like to highlight that ***our study covers many facets of distribution shifts***, including Dataset reproduction shift (e.g., ImageNet-V2 and CIFAR-10.2), style shift (e.g., ImageNet-R and WILDS-DomainNet) , blur shift (ImageNet-Blur), drastic visual variation (e.g., WILDS-iWildCam) and sketch shift (ImageNet-S). We note that some datasets are relatively near-ID, such as ImageNet-V2; some datasets are far-OOD (e.g., WILDS-iWildCam and ObjectNet).
>
> Last, we also thank the reviewer for mentioning Minderer et al 2021. This work systematically relates model calibration and accuracy. Based on this insight comment, ***we further discuss the post-hoc uncertainty calibration***, which does not require any network changes or re-training the network. For a perfectly calibrated model, the average of maximum Softmax prediction probability (MaxPred) over a test set corresponds to its accuracy over this dataset. However, calibration methods seldom exhibit desired calibration performance under distribution shifts (Ovadia et al., 2019). That said, it would be interesting to study post-hoc calibration methods for OOD datasets, which benefits MaxPred for assessing model generalization under distribution shift. We have added this discussion in Section 6.
>
> *Minderer et al. Revisiting the calibration of modern neural networks. In NeurIPS, 2021*
>
> *Ovadia et al. Can you trust your model’s uncertainty? evaluating predictive uncertainty under dataset shift. In NeurIPS, 2019*
>
> **Q2-2: It is possible that SoftmaxCorr by approximating the confusion matrix is simply a good proxy for ID test set accuracy or similar metrics and that these metrics and SoftmaxCorr will not generalize to far OOD datasets.**
>
> Thanks for this comment. We respectfully disagree that SoftmaxCorr is simply a good proxy for ID test set accuracy. We clarify that SoftmaxCorr captures models’ class-class confusion information on OOD datasets, and thus is effective in characterizing their OOD generalization ability. Moreover, when ID accuracy is not informative for OOD accuracy (please see the above Q1-1),  SoftmaxCorr still exhibits moderately high correlations. This also indicates that SoftmaxCorr is not a simple proxy for ID accuracy.
>
> **Q2-3: Evaluation on OOD datasets with label spaces different to the ID dataset. The SoftmaxCorr metric could be computed on OOD datasets that do not share a label space with the ID dataset. I would have liked the authors to have stress tested their method by evaluating it on datasets such as OOD=SVHN when ID=CIFAR-10 as done in Liu et al. (2020) and Van Amersfoort et al. (2020), and similarly more challenging label shifted datasets for the ID=ImageNet experiments.**
>
> Thanks for raising this point.  We would like to clarify that SoftmaxCorr is proposed for the task of ***OOD generalization assessment***, where the ***ID and OOD datasets share the same label space*** (Miller et al., 2021; Wenzel et al., 2022) and the goal is to characterize the impact of distribution shift. In comparison, the task of ***OOD data detection aims to find and reject the data from unseen/ open-set classes*** (as also mentioned by reviewer). Due to the ***significantly different nature of two tasks, we think SoftmaxCorr is not suitable for detecting unseen/ open-set data***. Similarly, some OOD detection methods would not be suitable for OOD generalization assessment. For example, the suggested two works as discussed above (Q1-2). In addition, we have newly tested two recent OOD detection methods (MaxLogit and Energy) and observe that they only give weak correlation (please refer to Q6/ Reviewer hesX for more details).
>
> We further note that when the label space of the test set is different from the training set (or ID dataset), the generalization problem does not hold. That said, studying datasets with different label spaces (e.g., ImageNet and SVHN) would beyond the scope of this work.
>
> In the revised version, we have highlighted the essential difference between OOD detection and OOD generalization.

---

> ### Author Response · Authors · 2022-11-17
> **Author response to Reviewer 8Aud (Part II)**
>
> **Q1-3:  A further interesting baseline, would be to use the confusion matrix directly (i.e., cosine similarity of the confusion matrix with the identity matrix) by labeling a small number of these samples … I am open to arguments as to why this may be impractical or undesirable in certain scenarios ... it would be useful to have this oracle and potentially practical method benchmarked in the experiments section.**
>
> Thanks for raising this Interesting idea. Following the suggestion, we have included this oracle. Specifically, we randomly sample 1%, 5% and 10% of labeled test data and compute the cosine similarity between the confusion matrix and identity matrix. We report the average of Spearman’s rank correlation over three random runs:
>
> |Dataset|1%| 5%|10%|
> | :---: | :---: | :---: | :---: |
> |ImageNet-R|0.984|0.993|0.993|
> |ImageNet-v2-A|0.947|0.981|0.990|
> |ImageNet-v2-B|0.857|0.956|0.974|
> |ImageNet-v2-C|0.922|0.979|0.986|
> |ImageNet-S|0.990|0.997|0.998|
> |Stylized-ImageNet|0.986|0.996|0.998|
> |ObjectNet|0.972|0.981|0.984|
> |ImageNet-Blur|0.989|0.993|0.995|
> |CIFAR-10.2|0.350|0.617|0.706|
> |CINIC|0.842|0.928|0.943|
> |CIFAR-10-Noise|0.842|0.950|0.956|
> |iWildCam-OOD|0.848|0.870|0.868|
> |Camelyon17-OOD|0.936|0.945|0.949|
> |DomainNet-OOD|0.955| 0.977| 0.968|
>
>
>
> This oracle baseline exhibits a strong correlation given 5% of labeled test data. This suggests that ***exploring class confusion information for OOD generalization assessment is feasible***. Specifically, if the calculated class-class correlation matrix ideally matches the true confusion matrix, then OOD generalization can be perfectly predicted.
>
> Furthermore, we would like to discuss that ***this oracle may be impractical*** in some scenarios. ***First***, data annotation could be very expensive and challenging. For example, precisely labeling wildlife categorization of iWildCam images is laborious and requires expert-level knowledge; annotating cancer tissue images of Camelyon17 also requires expert-level knowledge. ***Second***, when the test distribution is changed, we need to label data again which is also laborious.
>
> We have discussed this oracle baseline in the revised paper (Section A.6).
>
>
>
> **Q1-4: Comparison to non-softmax probability baselines.**
>
> A: In addition to the above analysis, we additionally include a baseline which is not softmax-based: model complexity (Jiang et al., 2018; Baldock et al., 2021). This is proposed to measure the complexity of models and has shown effective in ID generalization assessment (Jiang et al., 2020).
>
> Adversarial input margin (Complexity) is proposed to measure model complexity. It computes the smallest norm required for an adversarial perturbation in the input to change the model’s class predictions. It estimates adversarial input margin $\gamma$ by a linear approximation. Its formula is given by: $\gamma \simeq \min_{j \neq i} \frac{|z_{i} - z_{j}|}{|\triangledown_{x}(z_{i} - z_{j})|}$, where $x$ is the input, $i$ means predicted class and $z_j$ denotes the logit returned by the network for class $j$. We then take the average of $\gamma$ over all data.
>
> We report the Spearman's correlation ($\rho$) and weighted Kendall's correlation ($\rho$) of three methods and SoftmaxCorr on the following datasets: ImageNet-V2-A, ImageNet-S, CIFAR-10.2 and iWildCam-OOD. The new baseline (complext) exhibits weaker correlation than SoftmaxCorr. We have also discussed it in our revised paper (Section A.4)
>
> (1) Method comparison under $\rho$:
> |Dataset|Complexity|SoftmaxCorr|
> | :---: | :---: | :---: |
> |ImageNet-V2-A|0.341|0.922|
> |ImageNet-S|0.329|0.875|
> | CIFAR-10.2 |0.680|0.884|
>
> (2) Method comparison under $\tau_w$:
> |Dataset|Complexity|SoftmaxCorr|
> | :---: | :---: | :---: |
> |ImageNet-V2-A|0.282|0.792|
> |ImageNet-S|0.417|0.883|
> |CIFAR-10.2|0.524|0.694|
>
> *Baldock et al. Deep learning through the lens of example difficulty. In NeurIPS, 2021*
>
> *Jiang et al. Predicting the generalization gap in deep networks with margin distributions. In ICML, 2018*
>
> *Jiang et al. Neurips 2020 competition: Predicting generalization in deep learning*
>
> ***Based on the above discussion, we think that developing OOD measures based on softmax predictive probabilities is efficient and reasonable. Therefore, we mainly discuss softmax prediction-based methods and we view our correlation method (SoftmaxCorr) as a starting point. It would be promising to study other measures based on model weight and feature representations.***

---

> ### Author Response · Authors · 2022-11-17
> **Author response to Reviewer 8Aud (Part I)**
>
> **Q1-1: Comparison to non-softmax probability baselines. It would be understandable if the authors compared to other methods which may have additional requirements, and then stated that such methods have disadvantages due to these extra requirements. (1) It would have been useful to compare against methods which make use of ID performance, accuracy-on-the-line (Miller et al., 2021) and/or Wenzel et al. (2022).**
>
> Thank you for this valuable suggestion. Wenzel et al. (2022) report that in-distribution classification accuracy is the best predictor of OOD accuracy compared with other metrics (*e.g.*, in-distribution expected calibration error (ECE)). Therefore, we focus on the comparison with accuracy-on-the-line (Miller et al., 2021).
>
> In the following Table, we report Spearman's rank correlation ($\rho$) of accuracy-on-the-line and SoftmaxCorr.
>
> |Dataset| accuracy-on-the-line | SoftmaxCorr|
> | :---: | :---: | :---: |
> |ImageNet-R|0.932|0.925|
> |ImageNet-v2-A|0.995|0.922|
> |ImageNet-v2-B|0.994|0.912|
> |ImageNet-v2-C|0.995|0.917|
> |ImageNet-S|0.934|0.875|
> |Stylized-ImageNet|0.800|0.828|
> |ObjectNet|0.976|0.927|
> |ImageNet-Blur|0.902|0.967|
> |CIFAR-10.2|0.958|0.884|
> |CINIC|0.917|0.844|
> |CIFAR-10-Noise|0.003|0.813|
> |iWildCam-OOD|0.944|0.916|
> |Camelyon17-OOD|-0.021|0.689|
> |DomainNet-OOD|0.350|0.693|
>
> We observe that accuracy-on-the-line shows strong correlations on many datasets. However, we have similar observations as the authors of accuracy-on-the-line report: it exhibits weak correlation or even no correlation on CIFAR-10-Noise, Camelyon17-OOD and DomainNet-OOD ($\rho$ = 0.003, -0.021 and 0.350). In comparison, SoftmaxCorr remains relatively informative and exhibits moderately high correlation ($\rho$ = 0.813, 0.689 and 0.693) on the three test sets.
>
> We further note that when accuracy-on-the-line shows very strong correlations, SoftmaxCorr is also competitive. Therefore, we think that ***SoftmaxCorr is overall a good alternative when labeled ID data is inaccessible or accuracy-on-the-line does not hold***.
>
> Furthermore, we would like to highlight several ***benefits of SoftmaxCorr*** over accuracy-on-the-line. ***First***, we are unable to detect when accuracy-on-the-line gives very weak correlation because the process of identifying needs labeled OOD data. ***Second***, the access to ID labeled test data is not always guaranteed. Randomly splitting an ID test set from training data is a demanding process as well (Engstrom et al., 2020). ***Third***, since SoftmaxCorr does not require a held-out validation set,  we can use all the available data for training models.
>
> *Miller et al. Accuracy on the line: on the strong correlation between out-of-distribution and in-distribution generalization. In ICML, 2021*
>
> *Wenzel et al. Assaying out-of-distribution generalization in transfer learning. In  NeurIPS, 2022*
>
> *Engstrom et al. Identifying statistical bias in dataset replication. In ICML, 2020*
>
>
>
> **Q1-2: SNGP, Liu et al. (2020) and DUQ, Van Amersfoort et al. (2020). While they require architectural changes, it would be important to contextualize the SoftmaxCorr results by comparing against these methods.**
>
> Thank you for sharing the two studies of predictive uncertainty. We would like to first clarify that ***our work considers a significantly different task from the two researches***. Specifically, for the task of assessing out-of-distribution (OOD) generalization, we are provided with an OOD test set and various models and the goal is to develop a measure that can reflect and rank the generalization ability of models. Note that the ***OOD test set shares the same label space with the ID dataset***. For the task of predictive uncertainty, the goal is to make model predictions reflect the reliability or certainty for the input data. This can be used to find and reject abnormal data (e.g., data from unseen class).
>
> ***The suggested two works aim to achieve predictive uncertainty during training***. That is, they need to modify the model architecture as well as training objectives, and then re-train the model (also noted by the reviewer). If we use them for the task of assessing OOD generalization, we need to re-train all models with their methods. ***This might be impractical***, becases 1) it would be difficult to well optimize all networks (e.g., vision transformers) after modifying their architectures and loss functions. Also, training procedure needs a sophisticated hyper-parameter search. 2) After architecture modification and re-training, we are not assessing the original models of our interest. Therefore, we think the two suggested works ***are not suitable*** for the task of OOD generalization assessment. We have cited them in the revised version.
>
> *Liu, Jeremiah, et al. "Simple and principled uncertainty estimation with deterministic deep learning via distance awareness." In NeurIPS, 2020*
>
> *Van Amersfoort, Joost, et al. "Uncertainty estimation using a single deep deterministic neural network." In ICML, 2020*

---

### Author Response · Authors · 2022-12-05
**Common Response**

Dear Reviewers,

We deeply appreciate the insightful and constructive suggestions stemming from your thorough reviews. By addressing your concerns, we believe our work has been strengthened. In this common response, we refer to Reviewer 8Aud, Reviewer hesX and Reviewer Ku6q as R1, R2 and R3, respectively.

We study an ***important but under-explored task*** of characterizing OOD generalization. As acknowledged by R2 and R3, this task *“is important and has several practical applications from ML safety to adaptation” (R2)* and *“is an important problem” (R3)*. To this end, we propose a ***simple and effective OOD measure SoftmaxCorr***. It uses class-wise relationships encoded by the softmax prediction probabilities and considers both the certainty and diversity of model predictions. All reviewers point out that SoftmaxCorr is *“easy to compute and the underlying idea is simple and intuitive; novel; clear, simple and effective” (R1)*, *“simple and computationally inexpensive to derive” (R2)*, and *“novel; relatively simple”(R3)*. Furthermore, R2 and R3 commented that the effectiveness of SoftmaxCorr is validated with extensive analysis: *“thoroughly evaluated and performs consistently well” (R2)* and *“across a wide range of problems, models, and even training checkpoints, the proposed method shows strong empirical performance compared to well-established baselines” (R3)*.


Inspired by reviewers’ suggestions, we included more experimental analysis and discussions. We also have updated the main paper and appendix. We summarize the major changes below:

- Sensitivity analysis on test set size.  ***First***, we show that when the test set size is very small, all methods show slightly lower correlation strength. ***Second***, when test data are moderately sufficient, all methods have stable correlation (Q3-2/R1, Q3/R3, and Section 5.4 in the main paper).

- Compared with other methods. ***First***, we compare our method with three more methods: MaxLogit, Energy, and a non-softmax-based method, Adversarial input margin (Complexity). We observe that MaxLogit and Energy do not exhibit a good correlation with model generalization. While Complexity shows a weak correlation, it is still inferior to SoftmaxCorr (Q1-4/R1, Q6/R2, Section A.4 in the Appendix). ***Second***, we compare SoftmaxCorr with Accuracy-on-the-line. We show that SoftmaxCorr is overall a good alternative when labeled ID data is inaccessible or accuracy-on-the-line does not hold. Furthermore, compared with all methods, we emphasize that ***SoftmaxCorr is stable and consistently effective across different datasets*** (Q1-1/R1, Q5-1/R2, Section A.5 in the Appendix).

- Clarification on problem setting and far-OOD dataset. ***First***, we clarify the “far-OOD” dataset that has completely different classes is not suitable for studying OOD generalization. Please refer to Follow-Up Clarification (1/3) for R1. ***Second***, we thoroughly discuss that ***our problem setting is reasonable and practical*** from three aspects, please refer to Follow-Up Clarification (2/3) for R1.

- Clarification on contribution. We emphasize our major contribution and novelty in “Clarification on Contribution” of R2. Specifically, we explore a new application of Softmax prediction probability for characterizing model OOD generalization. We propose a simple yet effective measure named SoftmaxCorr and it is a non-trivial discovery. Furthermore, our new insight is that both prediction certainty and diversity are important and informative

We believe our work can inspire more research on OOD measures and thus is worth sharing with the community. We hope our reply addresses the questions of all reviewers. Moreover, we hope this common response is helpful for all reviewers and ACs in the next round of discussion. We remain available and eager to have a further discussion.

Kind regards,

Authors

---

### Decision · Program_Chairs · 2023-01-20

**Decision:**

Reject

**Justification For Why Not Higher Score:**

* Rejection unanimously supported by the reviewers.
* A lot of confusion about the goals and problem setting.
* Missing theoretical insights about the proposed criterion (why it works, when it is to break down, etc.)


**Justification For Why Not Lower Score:**

N/A

**Metareview: Summary, Strengths And Weaknesses:**

The reviewers and meta reviewer all carefully checked and discussed the rebuttal. They thank the authors for their response and their efforts during the rebuttal phase.
The reviewers and meta reviewer all acknowledge that the submission proposes a simple, yet surprisingly effective, way of capturing model out-of-distribution performance.
The response helped resolve some important concerns (e.g., strengthening of the experiments with additional baselines and sensitivity analysis).

At this stage, however, the submission still have some arguably important concerns that warrant further consolidations/investigations, for instance:

(i) The most important aspect is the fact that the approach is not theoretically grounded, which would help the readers to gain more insights as to why it works (and when it would break down—e.g., what happens with label shifts? Is the approach only suited for covariate shifts?). There is no doubt that the submission has done a massive experimental validation of the proposed criterion, but it remains difficult to build up intuition about the reported results.
Moreover, it seems that there are some features of the criterion not fully understood yet (e.g., SoftmaxCorr and “diag-sum” only differ from the scaling 1/(||C|| * ||I||); why 1/||C||, the only term with a dependence on the off-diagonal entries of C, has such an important effect?). Also, the fact that SoftmaxCorr does not properly support the case of a random classifier is problematic.

(ii) In the light of the increasingly dominant pre-training paradigm, it would be important to assess how pre-training affects SoftmaxCorr.

(iii) Another ablation study/sensitivity analysis that would be important to carry out is the effect of the train/val/test splits (see Section 6).

(iv) To further convince about the usefulness of SoftmaxCorr (and perhaps gain more insights), applications to early-stopping or hyperparameter tuning could be useful.

Because of the extremely competitive landscape of the submissions this year, the paper remains under the cut and has ultimately not been selected for acceptance.
We are convinced that the suggestions above will help strengthen the paper for a future resubmission, which the reviewers and meta reviewer all encourage.


**Summary Of Ac-Reviewer Meeting:**

The meta reviewer and two (out of the three reviewers) met on 12/6.

In the meeting, we discussed the following topics (that are reflected in the bullet points of the meta review):
* A lot of confusion about the actual problem setting and goal of the paper (e.g., if the goal is to compare models while ranking them, then it is unclear how to deal with the random classifier case); see (i).
* Missing insights, e.g., why the gap between SoftmaxCorr and diag-sum is so large and only explained by 1/||C||; see (i).
* Ways of gaining more insights; see (iv).
* Missing ablation studies; see (ii) and (iii).

The discussion led to a unanimous recommendation for rejection.